# Cell adhesion molecule KIRREL1 is a feedback regulator of Hippo signaling recruiting SAV1 to cell-cell contact sites

Atanu Paul [1], Stefano Annunziato[2], Bo Lu[1], Tianliang Sun[2], Olivera Evrova[2], Lara Planas-Paz[2], Vanessa Orsini[2], Luigi M. Terracciano[3,4], Olga Charlat[1], Zinger Yang Loureiro [1], Lei Ji[1], Raffaella Zamponi[1], Frederic Sigoillot [1], Hong Lei[1], Alicia Lindeman[1], Carsten Russ[1], John S. Reece-Hoyes[1], Thomas B. Nicholson [1], Jan S. Tchorz [2] & Feng Cong [1✉]

The Hippo/YAP pathway controls cell proliferation through sensing physical and spatial organization of cells. How cell-cell contact is sensed by Hippo signaling is poorly understood. Here, we identified the cell adhesion molecule KIRREL1 as an upstream positive regulator of the mammalian Hippo pathway. KIRREL1 physically interacts with SAV1 and recruits SAV1 to cell-cell contact sites. Consistent with the hypothesis that KIRREL1-mediated cell adhesion suppresses YAP activity, knockout of *KIRREL1* increases YAP activity in neighboring cells. Analyzing pan-cancer CRISPR proliferation screen data reveals KIRREL1 as the top plasma membrane protein showing strong correlation with known Hippo regulators, highlighting a critical role of KIRREL1 in regulating Hippo signaling and cell proliferation. During liver regeneration in mice, KIRREL1 is upregulated, and its genetic ablation enhances hepatic YAP activity, hepatocyte reprogramming and biliary epithelial cell proliferation. Our data suggest that KIRREL1 functions as a feedback regulator of the mammalian Hippo pathway through sensing cell-cell interaction and recruiting SAV1 to cell-cell contact sites.

---

[1] Novartis Institutes for BioMedical Research, Novartis Pharma AG, Cambridge, MA, USA. [2] Novartis Institutes for BioMedical Research, Novartis Pharma AG, Basel, Switzerland. [3] Department of Biomedical Sciences, Humanitas University, Pieve Emanuele (Milan), Italy. [4] IRCCS Humanitas Research Hospital, Rozzano (Milan), Italy. ✉email: feng.cong@novartis.com

First identified in *Drosophila*, the Hippo (Hpo) pathway has emerged as an evolutionarily conserved signaling pathway that regulates cell proliferation, tissue homeostasis, and tissue regeneration. Dysregulation of the Hippo pathway has been associated with aberrant tissue growth and tumorigenesis[1,2]. Central to this pathway is a kinase cassette consisting of MST1/2 (Hpo in fly) and LATS1/2 (Wts) along with their adapter proteins SAV1 (Sav) and MOB1A/1B (Mats), respectively. Once activated, the MST1/2-SAV1 complex phosphorylates and activates the LATS1/2-MOB1A/1B complex, which in turn phosphorylates and inactivates transcriptional co-activators YAP and TAZ (Yki in *Drosophila*) through limiting their nuclear translocation or promoting their degradation. When the kinase cascade is inactivated, unphosphorylated YAP/TAZ accumulate in the nucleus and bind with TEAD family transcription factor (Scalloped in *Drosophila*) to activate transcription of genes involved in cell proliferation, differentiation, and survival[3,4].

The Hippo pathway controls cell proliferation through sensing the physical state of cells within tissue. Hippo signaling mediates cell–cell contact-induced growth inhibition[5], but the exact mechanism by which cell–cell contact is sensed by Hippo pathway is not entirely clear. Hippo signaling is critical for tissue homeostasis and injury repair. Genetic ablation of core components of the Hippo pathway results in nuclear accumulation of YAP/TAZ, leading to tissue overgrowth and tumorigenesis. Aberrant activation of YAP/TAZ triggers a pro-growth transcriptional program conferring increased cell proliferation, migration, epithelial–mesenchymal transition (EMT), and cancer stem cell properties. Consistent with this, increased YAP/TAZ activity has been reported in a wide range of human malignancies, and often correlated with poor patient prognosis[2,6,7]. YAP/TAZ activity is also important for stem cell expansion and regeneration in response to tissue damage and inactivation of YAP/TAZ impairs tissue regeneration[8–10]. Therefore, dynamic and precise regulation of YAP/TAZ activity is essential for proper proliferative response during tissue homeostasis and regeneration.

While the core components of the Hippo pathway are highly conserved in *Drosophila* and mammals, recent molecular and genetic studies have demonstrated a clear divergence of upstream regulation of this pathway through evolution. Mammals appear to lack the functional homologs of many upstream regulators of *Drosophila* Hippo signaling. For the few with mammalian homologs, whether and how they regulate mammalian Hippo pathway is not entirely clear[11–14]. Findings that the plasma membrane functions as a crucial subcellular compartment to spatially organize the Hippo kinase cascade in both fly and mammals suggest that mammals likely have additional unknown Hippo regulators on the cell surface. In addition, while the Hippo pathway has been described as a linear pathway where the membrane-associated NF2 functions upstream of the MST1/2-SAV1 complex to activate LATS1/2 kinases, recent data suggest that NF2 and SAV1 function in parallel to recruit LATS1/2 and MST1/2 kinases, respectively, to the plasma membrane. This ensures spatial organization of LATS1/2 and its activating kinase (MST1/2) for optimal activation of the Hippo pathway[15]. However, the mechanism by which SAV1 itself is recruited to the plasma membrane in mammals remains unclear.

Using a genome-wide CRISPR LOF screen, we identified cell adhesion molecule KIRREL1 as an upstream regulator of the Hippo/YAP pathway, acting as a negative feedback regulator of YAP activity in vitro and in vivo. Mechanistically, KIRREL1 recruits SAV1 to cell–cell contact sites and enhances activation of Hippo pathway. Loss of KIRREL1 enhances YAP activity and increases cell proliferation at high density. Our study, therefore, suggests a potential mechanism of sensing cell–cell interaction by the mammalian Hippo pathway at the plasma membrane and provides insights into this key developmental pathway.

## Results

**KIRREL1 serves as a positive regulator of Hippo signaling.** CRISPR screen has emerged as a powerful tool to identify regulators of various signaling pathways. Here, we performed an unbiased genome-wide CRISPR screen in HEK293A cells to identify positive regulators of the Hippo pathway. We engineered HEK293A cells stably expressing Cas9 and GTIIC-GFP reporter, a transcription reporter of YAP/TEAD activity[16–18]. Since cells have decreased YAP activity when Hippo signaling is activated at high cell density, regular GFP cannot be used to track YAP activity due to its low turnover rate. We therefore fused a PEST sequence, which is found in many short-lived proteins, to the C-terminal of GFP to increase its turnover rate. The GFP^PEST variant has a comparatively short half-life allowing us to detect changes in YAP/TEAD activity. As expected, GTIIC-GFP^PEST reporter activity is regulated in a density-dependent manner corresponding to activated Hippo signaling at high cell density (Fig. S1a, S1b). For CRISPR screen, HEK293A GTIIC-GFP^PEST Cas9 cells were mutagenized by a pooled lentiviral guide RNA (gRNA) library, grown to high density, and subjected to fluorescence-activated cell sorting (FACS) analysis (Fig. 1a). Cells were at high density at the time of cell harvesting to increase the chance of identifying positive regulators of Hippo signaling. As shown in Fig. S1c, cells expressing low GFP or high GFP signal, respectively, were collected and subjected to next generation sequencing (NGS) and bioinformatics analysis. gRNAs overrepresented in the GFP-high population are expected to inhibit Hippo signaling and enhance GTIIC reporter activity. Known positive regulators of Hippo signaling, such as NF2, SAV1, TAOK1, VGLL4, and LATS1, scored clearly in the screen (Fig. 1b), validating the screening strategy. Interestingly, KIRREL1, a plasma membrane protein that has not previously been associated with Hippo signaling, scored as the second strongest hit of the screen, right after NF2 (Fig. 1b).

KIRREL1, also known as NEPH1 or KIRREL, is a transmembrane protein belonging to the Immunoglobulin superfamily (IgSF) of Cell Adhesion Molecules (CAMs), and it contains five immunoglobulin like domains in its extracellular region, a transmembrane domain, and an intracellular domain. To validate the screening results, two independent *KIRREL1* gRNAs were introduced into the HEK293A GTIIC-GFP^PEST Cas9 cell line by lentiviral transduction and pools of *KIRREL1* knockout (KO) cells were analyzed. *KIRREL1* KO increased GTIIC reporter activity (Figs. 1c and S1d) and expression of YAP/TAZ target genes, *CTGF, CYR61*, and *ANKRD1* (Fig. 1d). Consistent with the notion that TAZ protein is stabilized and YAP phosphorylation is decreased upon inactivation of Hippo signaling, *KIRREL1* knockout increased the expression of endogenous TAZ protein and decreased YAP phosphorylation in cells plated at high or medium density (Fig. 1e). Moreover, *KIRREL1* KO cells showed increased nuclear YAP expression when plated at high or medium density in nuclear fractionation assay (Fig. S1e). Using a parallel approach, we knocked down *KIRREL1* using two independent shRNAs in HEK293A GTIIC-GFP^PEST. Similar to CRISPR knockout data, knockdown of *KIRREL1* resulted in increased GTIIC-GFP^PEST reporter activity (Fig. 1f) and increased YAP/TAZ target gene expression (Fig. 1g). Taken together, these findings suggest that KIRREL1 is a positive regulator of the Hippo pathway.

**Overexpression of KIRREL1 limits YAP/TAZ activity.** As a complementary approach to loss-of-function experiments, we next investigated whether overexpression of KIRREL1 limits YAP/TAZ activity. In transient transfection experiments, over-expression of KIRREL1 decreased the expression of endogenous

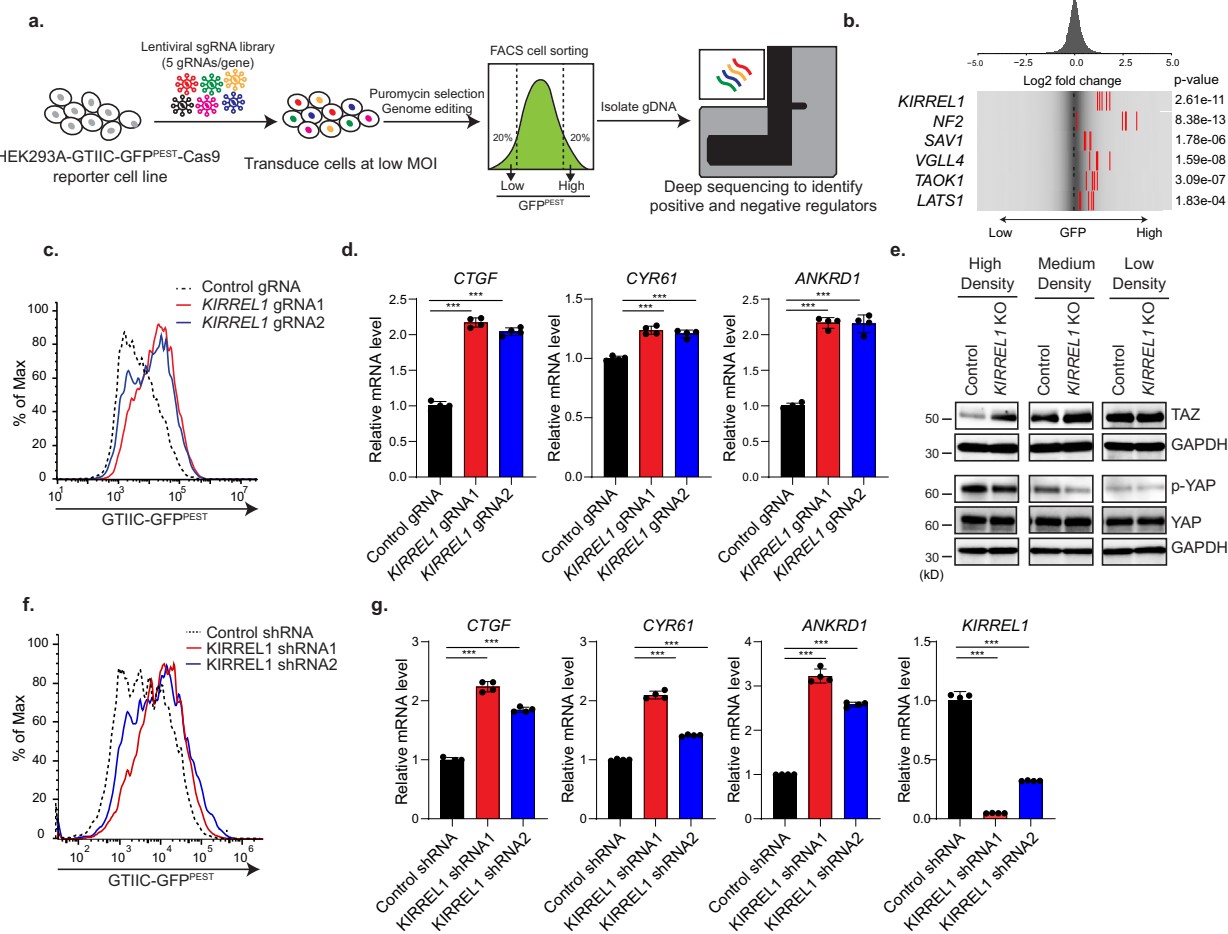

**Fig. 1 Genome-wide pooled CRISPR screen identified KIRREL1 as a positive regulator of Hippo pathway. a** Schematic of FACS-based pooled CRISPR screen to identify positive regulators of Hippo pathway in HEK293A GTIIC-GFP$^{PEST}$ Cas9 cells. **b** Frequency histograms of gRNAs identified in the high GFP$^{PEST}$ population of HEK293A GTIIC-GFP$^{PEST}$ pooled CRISPR screen. The RSA $p$-value is shown along with log2 fold change for each of the five gRNAs per gene for the indicated set of genes. gRNAs targeting the indicated genes are shown by the red lines. **c** Knockout of *KIRREL1* by independent gRNAs enhances GTIIC-GFP$^{PEST}$ reporter activity in FACS assay. HEK293A GTIIC-GFP$^{PEST}$ Cas9 cells transduced with lentivirus expressing RFP and control or *KIRREL1* gRNAs were mixed with parental cells at 1:3 ratio, co-cultured for 2 days, and subjected to FACS analysis. The signal of GTIIC-GFP$^{PEST}$ in RFP-positive cells is plotted. **d** Knockout of *KIRREL1* increases YAP target gene expression in HEK293A cells. HEK293A GTIIC-GFP$^{PEST}$ cells expressing control or *KIRREL1* gRNAs were plated at medium density ($0.5 \times 10^6$ cells/well) in 12-well plate and subjected for qRT-PCR analysis 48 h of post cell plating. The data represent mean ± SD, $n = 4$, error bars denote the SD between four biological replicates; Unpaired two-tailed $t$-test was used to determine the statistical significance ***$p$ value < 0.001. **e** Knockout of *KIRREL1* increases the protein level of TAZ and decreases YAP phosphorylation in HEK293A cells. Control and *KIRREL1* knockout cells were plated at high ($1 \times 10^6$ cells/well), medium ($0.5 \times 10^6$ cells/well), or low ($0.25 \times 10^6$ cells/well) density in 12-well plate and subjected to western blot analysis 24 h of post cell plating. GAPDH was used as loading control. **f** Depletion of *KIRREL1* using Dox-inducible shRNAs leads to increased GTIIC-GFP$^{PEST}$ reporter activity. HEK293A-GTIIC-GFP$^{PEST}$ cells stably expressing control or two independent Dox-inducible shRNAs targeting *KIRREL1* and subjected to FACS analysis following 96 h of Dox treatment. **g** Depletion of by shRNA increases expression of YAP target genes in HEK293A cells. The data represent mean ± SD, $n = 4$, error bars denote the SD between four biological replicates; Unpaired two-tailed $t$-test was used to determine the statistical significance, ***$p$ value < 0.001. Source data for Fig. 1d, e, g are provided as Source Data file.

TAZ protein (Fig. 2a), increased YAP phosphorylation (Fig. 2a), decreased YAP nuclear accumulation (Fig. S2a), and inhibited YAP-mediated GTIIC-luciferase reporter activity (Fig. S2b). Stable overexpression of KIRREL1 decreased GTIIC-GFP$^{PEST}$ reporter activity (Fig. 2b) and reduced the expression of YAP target genes *CTGF, CYR61, and ANKRD1* (Fig. 2c). The Immunoglobulin superfamily cell adhesion molecules (IgSF-CAMs) often associate with cytoskeletal or adapter proteins through their intracellular domain to trigger various intracellular signaling cascades[19,20]. This prompted us to probe whether KIRREL1-mediated activation of the Hippo pathway is dependent on its intracellular domain (ICD). To test this, we generated a KIRREL1 deletion mutant lacking the ICD (a.a. 521–757), hereafter referred to as ΔICD (Fig. 2d). Although the expression of the ΔICD

mutant was higher than full-length KIRREL1 as measured by western blot (Fig. S2c) and FACS analysis (Fig. S2d), overexpression of the KIRREL1 ΔICD mutant failed to inhibit YAP-mediated GTIIC-luciferase reporter activity (Fig. 2e). Together, these data suggest that KIRREL1 functions as a positive regulator of the Hippo pathway, and this activity requires its ICD.

**KIRREL1 regulates Hippo signaling by interacting with SAV1.** Having established that KIRREL1 functions as a positive regulator of the Hippo pathway, we next investigated how KIRREL1 is linked to the Hippo kinase cascade. To identify possible KIRREL1 interactors, we tested KIRREL1 binding with the core components of the Hippo pathway. We co-expressed Myc-KIRREL1-HA with Flag-tagged MST1, SAV1, LATS1, MOB1, NF2, and Kibra

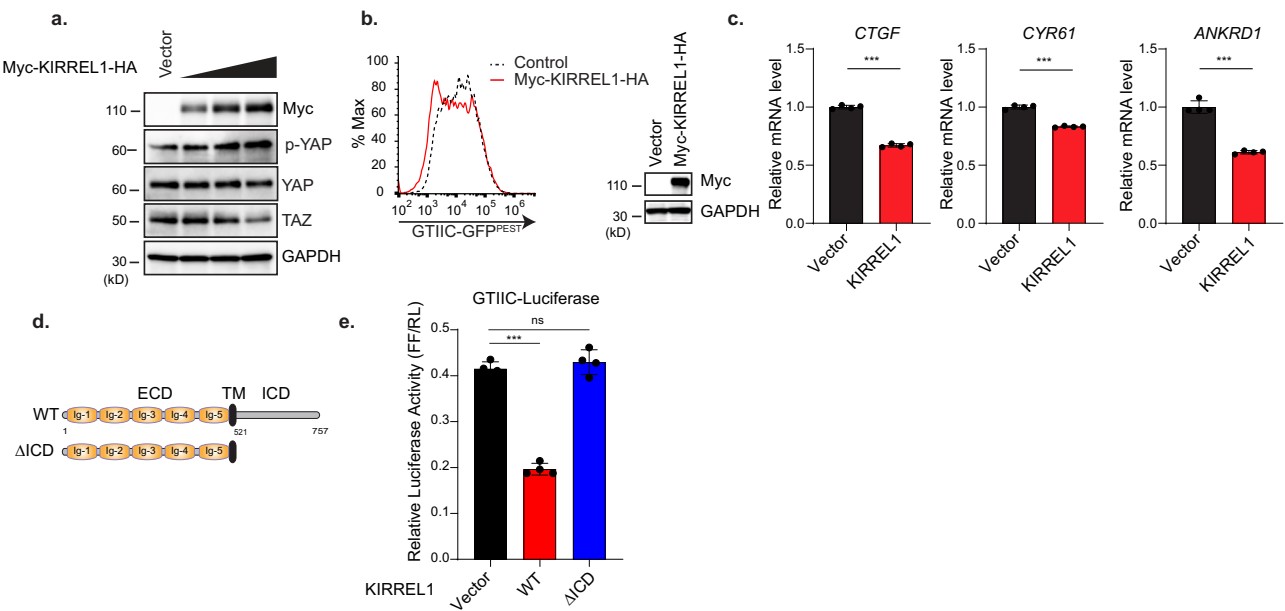

**Fig. 2 Overexpression of KIRREL1 enhances Hippo signaling in HEK293A cells. a** Overexpression of KIRREL1 decreases the expression of TAZ protein and increases YAP phosphorylation. HEK293A cells were transfected with increasing amount of Myc-KIRREL1-HA construct and seeded at medium density ($0.5 \times 10^6$ cells/well) in 12-well plate 24 h of post-transfection. Cells were harvested the next day and analyzed by western blot. **b** Left panel, overexpression of wild-type KIRREL1 decreases YAP-mediated GTIIC-GFP reporter activity in HEK293A GTIIC-GFP^PEST Cas9 cells. Right panel, western blot showing KIRREL1 expression. **c** Overexpression of wild-type KIRREL1 decreases YAP target gene expression. The data represent mean ± SD, $n = 4$, error bars denote the SD between four biological replicates; Unpaired two-tailed $t$-test was used to determine the statistical significance, ***$p$ value < 0.001. **d** Schematic of wild-type (WT) and ΔICD mutant of KIRREL1. **e** Overexpression of KIRREL1 inhibits YAP-responsive GTIIC-luciferase reporter activity. HEK293A cells were co-transfected with YAP and indicated constructs along with GTIIC-luciferase, and TK-*Renilla* luciferase, and subjected to dual luciferase assay. The data represent mean ± SD, $n = 4$, error bars denote the SD between four biological replicates; Unpaired two-tailed $t$-test was used to determine the statistical significance biological replicates, ***$p$ value < 0.001, ns—0.3886. Source data for Fig. 2a–c, e are provided as Source Data file.

and performed co-immunoprecipitation experiments. As shown in Fig. 3a, we detected a clear interaction with SAV1, marginal interaction with LATS1 and NF2, and no interaction with MST1, MOB1, and Kibra. Consistent with the finding that the KIRREL1 ICD is essential for inhibiting YAP/TAZ activity (Fig. 2e), deletion of the KIRREL1 ICD abrogated the binding between KIR-REL1 and SAV1 (Fig. 3b).

We then sought to determine which region of the KIRREL1 ICD is responsible for binding SAV1. The KIRREL1 ICD lacks any recognizable protein domain. We generated KIRREL1 ICD deletion mutants with the N-terminal (a.a 521–600—ICD Δ1), central (a.a 601–680—ICD Δ2), or C-terminal (a.a 681–757—ICD Δ3) regions of the ICD deleted (Fig. 3c). FACS analysis confirmed that these deletion mutants were expressed on the plasma membrane (Fig. S3a). Co-immunoprecipitation experiments revealed that while ICD Δ1 and ICD Δ2 mutants efficiently interacted with SAV1 similar to full-length KIRREL1, deletion of the C-terminal end of the KIRREL1 ICD (a.a. 681–757—ICD Δ3) completely abolished KIRREL1–SAV1 interaction (Fig. 3d). These results suggest that the C-terminal end of the KIRREL1 ICD is required for interacting with SAV1. We further tested whether the C-terminal region of the KIRREL1 ICD is sufficient to mediate interaction with SAV1. To this end, we purified a recombinant His-SUMO-tagged C-terminal fragment (a.a. 681–757—ICD$_{77}$ C-ter) or N-terminal fragment (a.a. 521–600—ICD$_{80}$ N-ter) of the KIRREL1 ICD and performed a Ni-NTA pull-down assay by mixing purified KIRREL1 ICD fragments with cell lysates expressing Flag-tagged MST1 or SAV1. As shown in Fig. 3e, the purified C-terminal fragment of the KIRREL1 ICD (ICD$_{77}$ C-ter), but not the N-terminal fragment of the KIRREL1 ICD (ICD$_{80}$ N-ter), bound to SAV1. We did not observe any

interaction between the purified KIRREL1 ICD and MST1 (Fig. 3e—right panel), indicating the specificity of KIRREL1 ICD-SAV1 interaction. In addition, we performed a Ni-NTA pull-down assay by mixing the recombinant ICD$_{77}$ C-ter or ICD$_{80}$ N-ter of KIRREL1 ICD and purified Flag-tagged SAV1. ICD$_{77}$ C-ter, but not ICD$_{80}$ N-ter, pulled down purified SAV1 (Fig. 3f), suggesting that the C-terminal end of the KIRREL1 ICD directly binds to SAV1. To test whether the C-terminal region of the KIRREL1 ICD is essential for inhibiting YAP activity, we performed a GTIIC-luciferase assay in HEK293A cells. As expected, while overexpression of wild-type (WT) KIRREL1 and the ICD Δ1 mutant inhibited YAP-mediated GTIIC-luciferase reporter activity, deletion of the C-terminal end of the KIRREL1 ICD (ICD Δ3) abrogated this inhibitory effect (Fig. 3g and Fig. S3b). Consistently, overexpression of WT and ICD Δ1, but not ICD Δ3, reduced YAP target gene expression (Fig. S3c) and decreased expression of endogenous TAZ protein (Fig. S3d).

We next investigated which domain of SAV1 is required for KIRREL1 binding. SAV1 contains an N-terminal flexible domain, two tandem WW domains, and a C-terminal SARAH domain (Fig. 3h). The SARAH domain of SAV1 binds to the SARAH domain of MST1/2 to form a heterodimer, and the WW domains of SAV1 homodimerize to promote the formation of MST-SAV1 heterotetramer. Heterotetramerization of MST-SAV1 complex enables trans-autophosphorylation and activation of MST kinase[21,22]. However, the function of the N-terminal flexible domain remains elusive. To identify the domain mediating binding with KIRREL1, we generated deletion mutants of SAV1 (Fig. 3h) and tested their binding to KIRREL1 in co-immunoprecipitation experiments. Interestingly, neither the

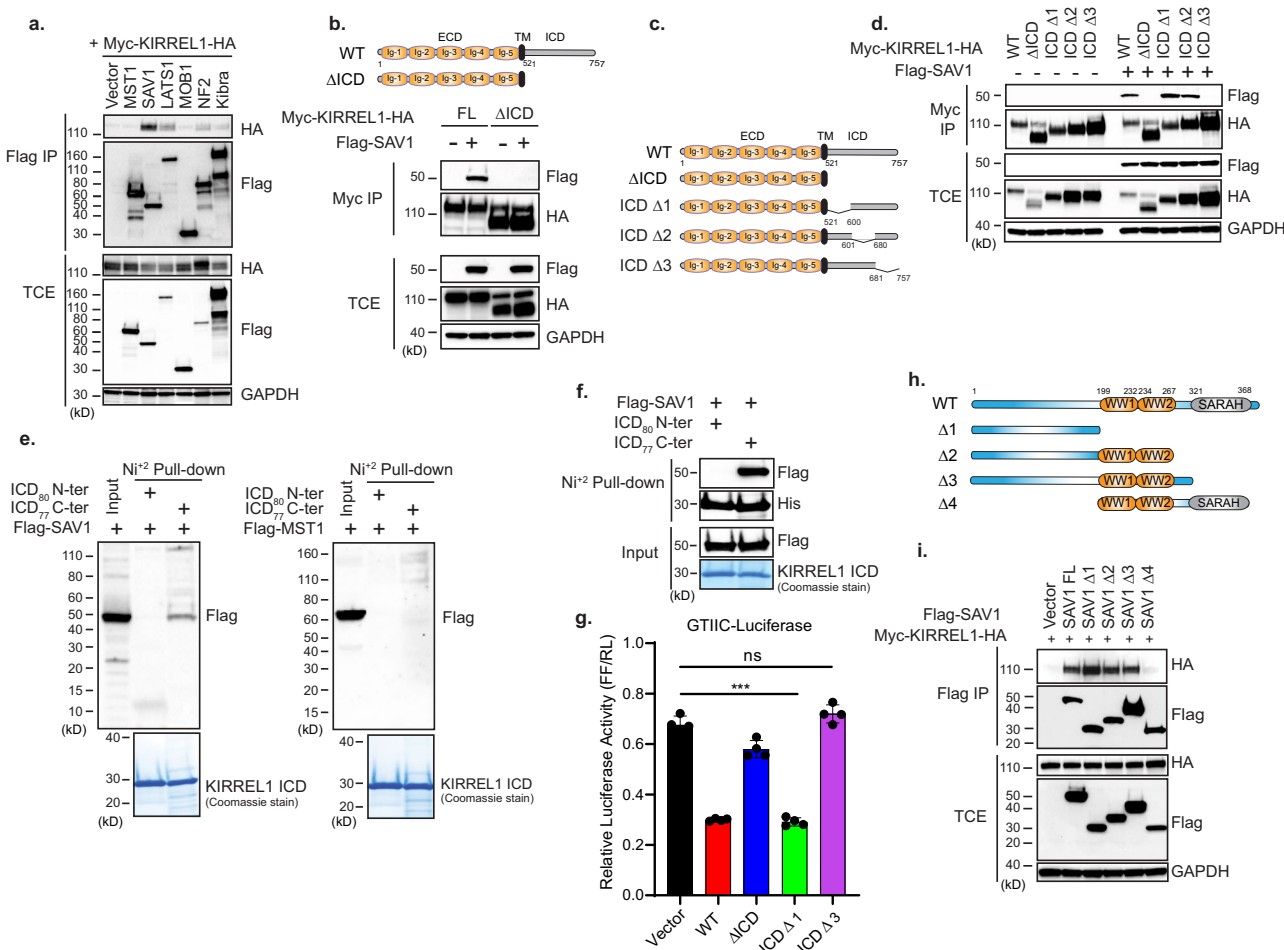

**Fig. 3 KIRREL1 interacts with N-terminal of SAV1 through its intracellular domain (ICD). a** KIRREL1 interacts with SAV1 in co-immunoprecipitation assay. HEK293T cells were co-transfected with plasmids encoding Myc-KIRREL1-HA and Flag-tagged Hippo pathway components, and cell lysates were subjected to co-immunoprecipitation assay. TCE total cell extract. **b** Full-length KIRREL1 but not the ΔICD mutant interacts with SAV1 in co-immunoprecipitation assay. **c** Schematic of full-length and ICD deletion mutants of KIRREL1. **d** Full-length, ICD Δ1, and ICD Δ2 but not ICD Δ3 mutant of KIRREL1 interacts with SAV1 in co-immunoprecipitation assay. **e** The C-terminal fragment of KIRREL1 ICD interacts with SAV1. Purified recombinant His-SUMO-tagged KIRREL1 N-terminal fragment of ICD (ICD$_{80}$—a.a 521–600) or C-terminal fragment of ICD (ICD$_{77}$—a.a 681–757) was incubated with cell lysates expressing ectopic Flag-tagged SAV1 (left panel) or MST1 (right panel) and subjected to Ni-NTA pull-down. Bottom panel shows Coomassie-stained gel of purified KIRREL1 ICD fragments. **f** The C-terminal fragment of KIRREL1 ICD directly interacts with SAV1. Purified recombinant His-SUMO-tagged KIRREL1 N-terminal fragment of ICD (ICD$_{80}$—a.a 521–600) or C-terminal fragment of ICD (ICD$_{77}$—a.a 681–757) was incubated with purified Flag-SAV1 and subjected to Ni-NTA pull-down assay. Coomassie-stained gel shows purified HIS-SUMO-tagged KIRREL1 ICD fragments. **g** Overexpression of KIRREL1-induced inhibition of GTIIC-luciferase activity requires the C-terminal fragment of KIRREL1 ICD. Experiment was performed as in Fig. 2e. The data represent mean ± SD, $n = 4$, error bars denote the SD between four biological replicates; unpaired two-tailed $t$-test was used to determine the statistical significance, ***$p$ value < 0.001, ns—0.1346. **h** Schematic of full-length and deletion mutants of SAV1. **i** N-terminal flexible domain of SAV1 is required and sufficient for binding with KIRREL1. HEK293T cells were transfected with plasmids encoding KIRREL1 and indicated SAV1 mutants and subjected to co-immunoprecipitation assay. Source data for Fig. 3a, b, d, e, f, g, i are provided as Source Data file.

WW nor the SARAH domain is required for binding with KIRREL1, while the N-terminal flexible domain is required and sufficient for KIRREL1 binding (Fig. 3i). Taken together, these results suggest that KIRREL1 functionally interacts with SAV1 through its C-terminal end of the ICD and the N-terminal flexible domain of SAV1.

**KIRREL1 and SAV1 function in the same molecular axis.** Our findings suggest that KIRREL1 activates Hippo signaling through interacting with SAV1. To further test this hypothesis, we inhibited KIRREL1 and SAV1 individually or in combination and measured YAP activity. If KIRREL1 and SAV1 function through different axes in modulating Hippo signaling, inhibition of KIRREL1 and SAV1 would likely have additive effects on YAP activity. In contrast, if KIRREL1 and SAV1 act in the same signaling axis, inhibition of KIRREL1 and SAV1 should not have additive effect on YAP activity. We introduced control or *SAV1* gRNAs into HEK293A GTIIC-GFP$^{PEST}$Cas9 cells stably expressing Dox-inducible control shRNA or KIRREL1 shRNA to inhibit either KIRREL1 or SAV1, or both. Western blot and quantitative RT-PCR confirmed *SAV1* KO and *KIRREL1* knockdown, respectively (Fig. S4a, S4b). Inhibition of KIRREL1 or SAV1 alone increased GTIIC reporter activity to the same degree (Fig. 4a). Notably, co-inhibition of KIRREL1 and SAV1 did not further increase GTIIC reporter activity (Fig. 4a) and expression of YAP

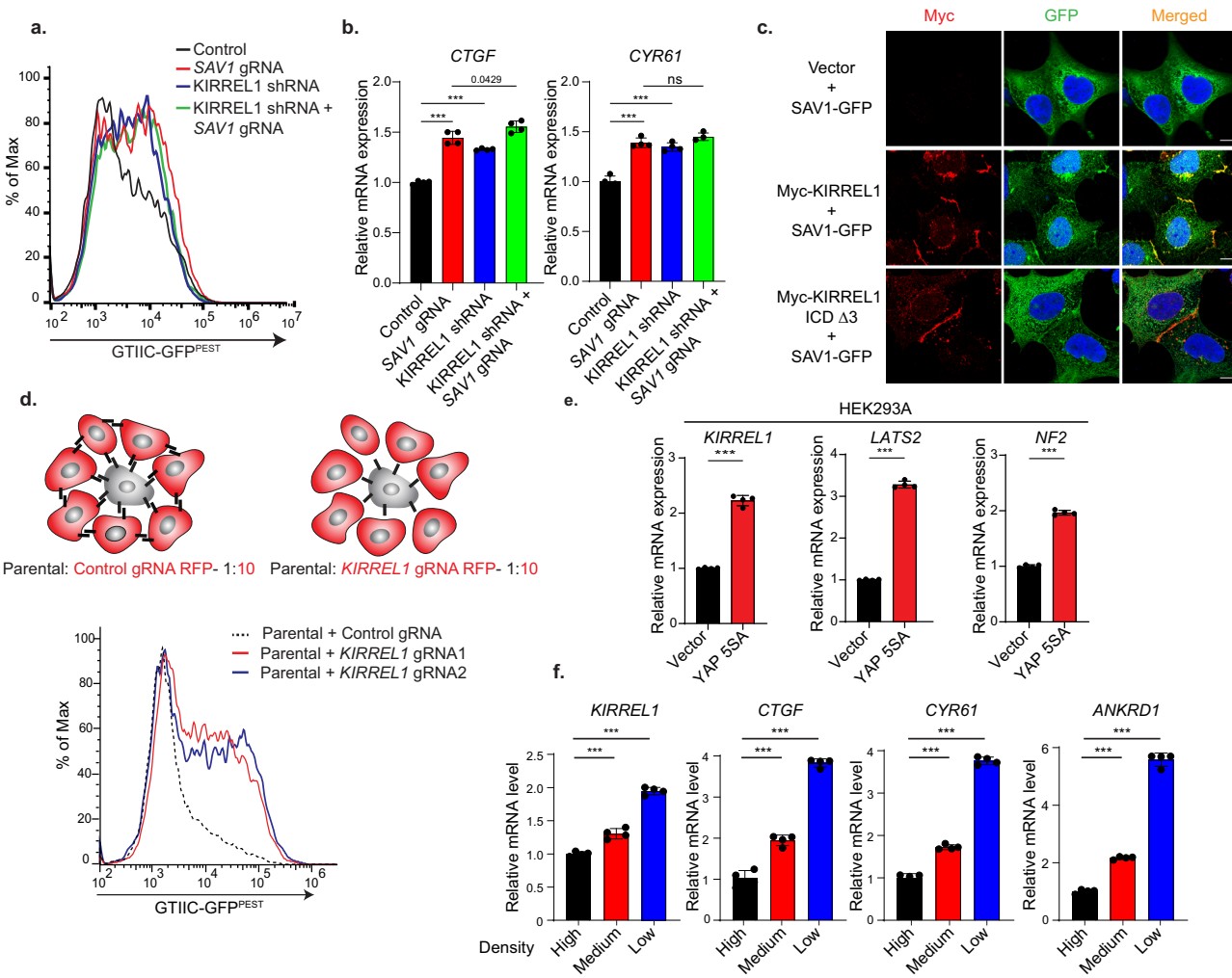

**Fig. 4 KIRREL1 regulates Hippo signaling through recruiting SAV1 to the cell–cell contact sites. a** Dual inhibition of *KIRREL1* and *SAV1* has no additive effect on GTIIC-GFP[PEST] reporter. HEK293A GTIIC-GFP[PEST] Cas9 cells with inhibition of *KIRREL1*, *SAV1* or both were subjected to flow cytometry analysis. **b** Dual depletion of *KIRREL1* and *SAV1* has no additive effect on expression of YAP target genes. Cells were plated at medium density ($0.5 \times 10^6$ cells/well) in 12-well plate and subjected to qRT-PCR analysis 48 h of post cell plating. Data represent mean ± SD, $n = 4$, error bars denote the SD between four biological replicates; Unpaired two-tailed *t*-test was used to determine the statistical significance, \*\*\**p* value < 0.001, ns—0.0978. **c** Overexpression of KIRREL1 promotes localization of SAV1 to the cell-cell contacts. HEK293A cells were co-transfected with plasmids encoding GFP-tagged SAV1 and Myc-tagged KIRREL1 and subjected to immunofluorescence assay using anti-Myc and anti-GFP antibodies. DAPI was used to mark the nuclei. Scale bar 10 μm. **d** Knockout of *KIRREL1* enhances YAP activity in neighboring cells. HEK293A GTIIC-GFP[PEST] Cas9 cells and HEK293A GTIIC-GFP[PEST] Cas9 cells transduced with lentivirus expressing RFP and control or *KIRREL1* gRNAs were mixed at 1:10 ratio, co-cultured for 2 days, and subjected to FACS analysis. The signal of GTIIC-GFP[PEST] in RFP-negative parental cells is plotted**. e** Active YAP induces expression of *KIRREL1* mRNA. HEK293A cells stably expressing vector or YAP 5SA mutant were plated at medium density ($0.5 \times 10^6$ cells/well) in 12-well plate and subjected to qRT-PCR analysis 48 h of post cell plating. *LATS2* and *NF2* were used as positive control. The data represent mean ± SD, $n = 4$, error bars denote the SD between four biological replicates; Unpaired two-tailed *t*-test was used to determine the statistical significance, \*\*\**p* value < 0.001. **f** *KIRREL1* mRNA expression is regulated by cell density. HEK293A cells were plated at high ($1 \times 10^6$ cells/well), medium ($0.5 \times 10^6$ cells/well), or low density ($0.25 \times 10^6$ cells/well) in 12-well plate and subjected to qRT-PCR analysis 48 h of post cell plating. *CTGF, CYR61*, and *ANKRD1* were used as positive control. The data represent mean ± SD, $n = 4$, error bars denote the SD between four biological replicates; Unpaired two-tailed *t*-test was used to determine the statistical significance. Source data for Fig. 4b, e, f are provided as Source Data file.

target genes *CTGF* and *CYR61* (Fig. 4b). Together, these findings are in favor of the hypothesis that KIRREL1 and SAV1 function in the same molecular axis to modulate Hippo signaling.

**KIRREL1 promotes SAV1 localization at the cell–cell contacts.** We next sought to investigate the mechanism by which KIRREL1–SAV1 interaction modulates the Hippo pathway. Previous studies suggest that KIRREL1 undergoes trans-dimerization and mediates cell–cell interaction of kidney podocytes[23–26]. Moreover, recent findings suggest that membrane SAV1 recruits

MST kinases to the plasma membrane to phosphorylate membrane-localized LATS kinases[15]. However, how SAV1 itself is recruited to the plasma membrane in mammals remains unclear. Since KIRREL1 directly binds to SAV1, we tested whether KIRREL1 recruits SAV1 to cell–cell contacts by coexpressing GFP-tagged SAV1 with full-length KIRREL1 or a KIRREL1 ICD Δ3 mutant. Consistent with the notion that KIRREL1 mediates cell–cell interaction, KIRREL1 was highly enriched at the cell–cell contacts and barely detected on the cell surface that does not interact with neighboring cells (Fig. 4c). Significantly, although SAV1 showed minimal localization at cell–cell contact sites in

control cells, SAV1 was predominantly localized at cell–cell contact sites when coexpressed with KIRREL1, co-localizing with KIRREL1 (Fig. 4c). Consistent with the finding that the C-terminal end of the KIRREL1 ICD is responsible for SAV1 binding, SAV1 showed minimal localization to the cell–cell contact sites when co-expressed with KIRREL1 ICD Δ3 mutant (Fig. 4c). Together, these findings suggest that KIRREL1 promotes recruitment of SAV1 to cell–cell contact sites to activate the downstream kinase cassette, linking high cell density to the activation of the Hippo pathway.

**KIRREL1 inhibits YAP activity in neighboring cells.** Since KIRREL1 mediates cell–cell adhesion through its extracellular domain, we hypothesized that KIRREL1 is required for cell–cell adhesion-mediated regulation of Hippo/YAP signaling. This hypothesis predicts that knockout of *KIRREL1* increases YAP activity not only in cells lacking KIRREL1, but also in neighboring cells expressing wild-type KIRREL1. To test this, HEK293A GTIIC-GFP[PEST] Cas9 cells (RFP negative parental) and HEK293A GTIIC-GFP[PEST] cells transduced with lentivirus expressing control or *KIRREL1* gRNAs (RFP positive) were mixed at 1:10 ratio, co-cultured for 2 days, and analyzed by FACS assay. Knockout of *KIRREL1* significantly increased YAP activity in RFP-negative parental cells expressing wild-type KIRREL1 (Fig. 4d and Fig. S4c), which strongly supports the hypothesis that KIRREL1-mediated cell adhesion suppresses YAP activity.

**YAP increases *KIRREL1* expression as a feedback mechanism.** Signaling pathways often contain negative feedback loops where the expression of pathway components is regulated by the pathway itself to fine-tune the signal output. Hippo signaling shows remarkable negative feedback regulation; expressing an activating mutant of YAP in vivo leads to compensatory increase of YAP phosphorylation[27], and expression of both *NF2* and *LATS2* is induced by YAP activation[28,29]. Given KIRREL1's role in inhibiting YAP activity, we tested whether YAP might induce *KIRREL1* expression to establish a negative feedback loop. We found that stable expression of YAP 5SA, a constitutively active YAP mutant with all five LATS phosphorylation sites mutated[5], in HEK293A or U251-MG cells not only increased the expression of *LATS2* and *NF2*, but also increased the expression of *KIRREL1* (Figs. 4e and S4d). Conversely, double KO of *YAP* and *TAZ* decreased the expression of *KIRREL1* (Fig. S4e). Consistent with the notion that YAP activity is regulated in a cell density-dependent manner[5], *KIRREL1* expression was high at low cell density and low at high cell density, similar to known YAP target genes *CTGF, CYR61,* and *ANKRD1* (Fig. 4f). Collectively, these findings suggest that YAP activity triggers increased expression of *KIRREL1* to activate Hippo signaling as a feedback mechanism to restrict YAP activity.

**KIRREL1 loss enhances YAP-mediated cancer cell proliferation.** Based on the function of KIRREL1 in inhibiting YAP/TAZ activity, we speculated that knockout of *KIRREL1* would lead to increased cell proliferation. To this end, we took an unbiased approach by analyzing datasets from the Cancer Dependency Map project (DepMap portal—https://depmap.org/portal). The DepMap is an ongoing project to systematically assess the effect of single-gene inactivation on cell proliferation by genome-wide RNAi or CRISPR screens across a large panel of well-characterized human cancer cell lines. The essentiality of the gene for cell proliferation is represented as a dependency score—denoted as CERES score for CRISPR screens (Project Achilles)[30,31]. Positive or negative CERES scores suggest either increased or decreased proliferation, respectively, upon gene KO.

Using these large publicly available datasets, we analyzed the effect of *KIRREL1* knockout in more than 700 cancer cell lines (CRISPR Avana Public 20Q3). First, since correlation between dependency profiles often suggests their functionality in the same pathway, we assessed the co-dependency relationship of *KIRREL1* with other genes. Strikingly, *KIRREL1* is strongly associated with multiple known Hippo regulators including *SAV1* (Fig. 5a). In fact, *KIRREL1* is the second most correlated gene for both *SAV1* and *NF2* (Fig. 5b, c), and knockout of *KIRREL1, NF2,* or *SAV1* increased proliferation of a similar group of cell lines (Fig. S5a, S5b). Of note, *KIRREL2* and *KIRREL3* did not score with any Hippo pathway regulators in these correlation studies (Fig. S5c, S5d), consistent with their restricted expression pattern[32] compared to *KIRREL1* that is expressed in a wide variety of tissues (Fig. S5e). The fact that KIRREL1 scores as the top plasma membrane protein with the strongest correlation with known Hippo regulators across multiple cell lines emphasizes the critical role of KIRREL1 in regulating Hippo signaling and cell proliferation at the cell surface.

Next, we plotted *KIRREL1* mRNA expression over the *KIRREL1* dependency score (CERES) for over 700 cancer cell lines. A group of cell lines, all with medium to high expression of *KIRREL1*, have a clear positive CERES score, suggesting that KO of *KIRREL1* in these lines increases cell proliferation (Fig. 5d). We selected several cancer cell lines including NCI-H2030 (lung adenocarcinoma), IGR-39 (melanoma), GAMG (glioblastoma), and DAOY (medulloblastoma) for validation. DAOY was selected based on pooled shRNA proliferation screening data[33]. *KIRREL1* knockout led to a significant increase in the expression of YAP target genes *CTGF, CYR61,* and *ANKRD1* (Fig. 5e), suggesting that KIRREL1 restricts YAP/TAZ activity in these cancer cell lines. In addition, knockout of *KIRREL1* in IGR-39 cells decreased phospho-YAP and increased TAZ protein expression indicating enhanced YAP/TAZ activity upon KIRREL1 depletion (Fig. S5f). Consistent with enhanced YAP/TAZ activity, knockout of *KIRREL1* in NCI-H2030, IGR-39, GAMG, and DAOY increased cell proliferation (Fig. 5f). Notably, while the proliferative advantage of KIRREL1 knockout cells was evident at high density at later time points in the proliferation assay, this effect was less obvious at early time points when cells were at low density. Since KIRREL1 mediates cell-cell adhesion through trans-dimerization of its extracellular domain[24,25,34,35], KIRREL1 might serve as a sensor for cell-cell contact in high cell density-mediated activation of the Hippo pathway and inhibition of cell proliferation.

**KIRREL1 restricts YAP/TAZ activity during ductular reaction.** To investigate KIRREL1-mediated regulation of Hippo signaling in vivo, we sought to examine the function of KIRREL1 in a physiological process where Hippo-YAP signaling is known to play a critical role. To this end, we assessed the function of KIRREL1 during the liver damage-induced hepatic ductular reaction (DR). DR is a regenerative process during which biliary epithelial cells (BECs) are activated and expand following hepatic injury around the portal vein, to form a transient luminal epithelium and establish an auxiliary biliary system. We and others have recently shown that YAP/TAZ activity plays a crucial role in 3,5-Dicarbethoxy-1,4-dihydrocollidine (DDC)-induced DR[36–38]. In naive adult C57Bl/6 wild type (WT) control mouse livers, *Kirrel1* mRNA was almost exclusively expressed in a subset of CK19+ biliary epithelial cells (BECs) and a small fraction of periportal hepatocytes with no expression around the central vein as indicated by ISH staining for *Kirrel1* mRNA (Figs. 6a and S6a). In response to 16 days DDC diet, control livers showed a marked increase in *Kirrel1* ISH signals in both CK19 + BECs comprising

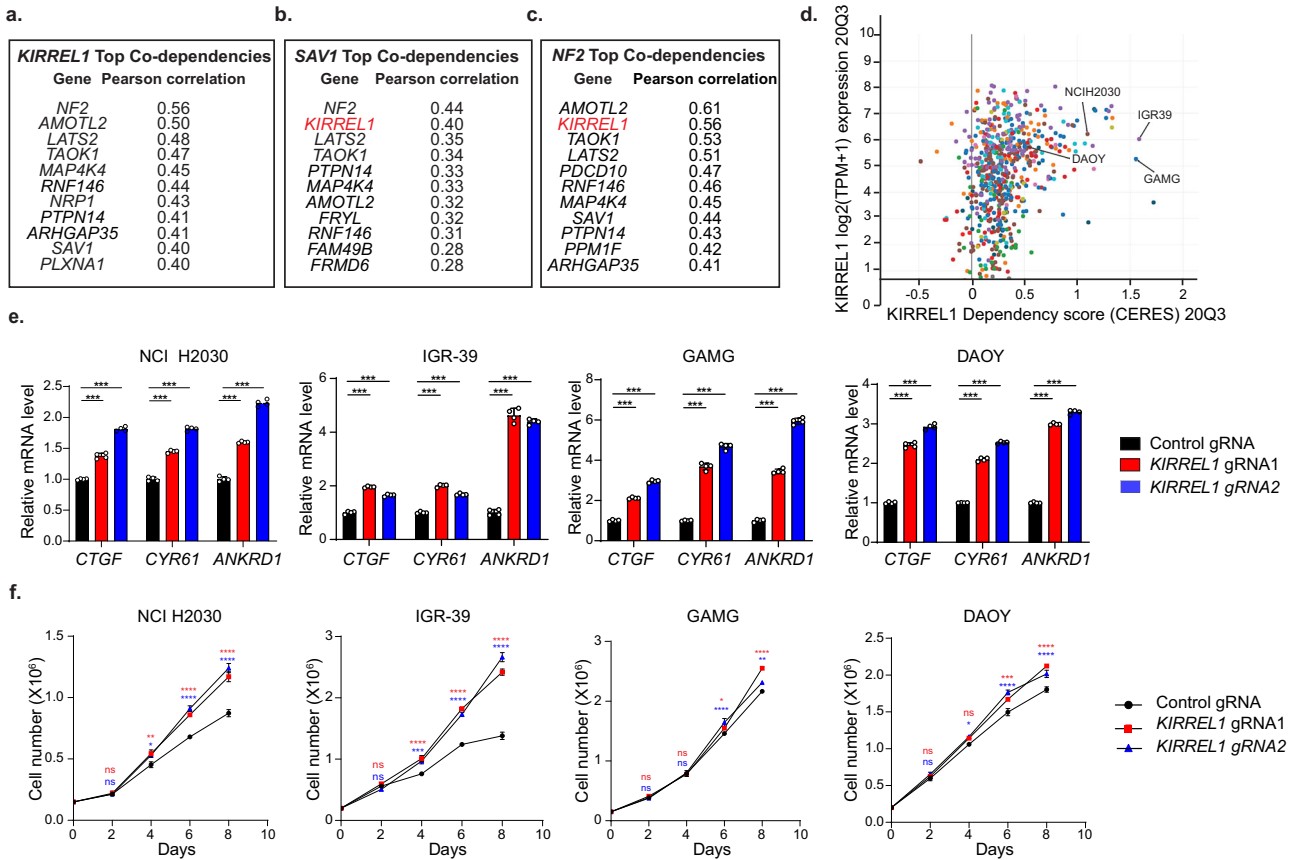

**Fig. 5 DepMap analysis reveals co-dependency of *KIRREL1* and known Hippo pathway genes in CRISPR proliferation screens. a** Top *KIRREL1* correlated genes in Project Achilles. Pearson correlation score on the right indicates the strength of co-dependency between *KIRREL1* and indicated genes. **b** Top *SAV1* correlated genes in Project Achilles. **c** Top *NF2* correlated genes in Project Achilles. **d** Dot plot analysis of cancer cell lines with *KIRREL1* mRNA expression (*y*-axis) plotted against *KIRREL1* dependency score (*x*-axis). High CERES score suggests increased cell proliferation upon gene knockout. **e** Knockout of *KIRREL1* increases expression of YAP target genes in cancer cell lines. Data represent mean ± SD, n = 4, error bars denote the SD between four biological replicates; Unpaired two-tailed *t*-test was used to determine the statistical significance ***p value < 0.001. **f** Knockout of *KIRREL1* promotes proliferation of cancer cell lines. Cancer cell lines expressing control or *KIRREL1* gRNAs were plated in 12-well plates, and cell numbers were quantified at indicated time-points. Data represent mean ± SD, n = 3, error bars denote the SD between three replicates; two-way ANOVA was used without any adjustments to determine the statistical significance. ns—p value > 0.05, *p value < 0.05, **p value < 0.01, ***p value < 0.001. The exact p values are provided as source data. Source data for Fig. 5e, f are provided as Source Data file.

the DR as well as in periportal hepatocytes (Figs. 6b, c), consistent with basal YAP/TAZ activity in BECs of naive mice and marked increase of YAP/TAZ activity in BECs and periportal hepatocytes following DDC, respectively[36–38]. These findings are consistent with induction of *KIRREL1* expression by YAP in cell culture (Fig. 4e) and suggest that DDC diet-induced YAP stimulates *Kirrel1* expression as a feedback mechanism. Next, to study the functional role of *Kirrel1* during DR in mice, we generated mice with *Kirrel1* deletion in BECs and hepatocytes (Kirrel1:AlbCre mice) (Fig. S6b). The absence of *Kirrel1* ISH signals in the liver (Fig. 6a) but robust expression in kidney glomeruli (Fig. S6c) confirmed successful hepatic deletion of endogenous *Kirrel1*. Kirrel1:AlbCre mice showed no overt phenotype and displayed normal liver development as indicated by intact liver zonation and architecture (Fig. S6d). However, in response to 16 days DDC diet, Kirrel1:AlbCre livers showed a substantial increase in *Ctgf* ISH signal compared to control livers (Fig. 6d), suggesting increased YAP/TAZ activity during liver regeneration in Kirrel1:AlbCre mice. Consequently, Kirrel1:AlbCre mice showed a stronger DR response compared to control mice as indicated by increased numbers of CK19 + BECs and SOX9+ reprogrammed periportal hepatocytes (Fig. 6e–g and S6e). In addition, Kirrel1:AlbCre mice showed a substantial increase in proliferating

CK19 + BECs (Fig. 6h, i). Consistently, YAP activity was shown to mediate a DR by promoting both proliferation of CK19 + BECs as well as upregulation of SOX9 in periportal hepatocytes, which then transdifferentiate into BECs[36–38]. Similar liver-to-body weight ratio, liver serum markers and inflammation in response to DDC diet suggest comparable injury in Kirrel1:AlbCre mice and controls (Fig. S7a–d). Interestingly, human livers with a DR show a high correlation between ISH signals for *KIRREL1* and the YAP target gene *KLF6*, with high levels of expression in ductular reaction areas and lower expression in the liver parenchyma (Fig. S7e, f). This suggests that KIRREL1 may also play a role in restricting YAP activity during a DR in human livers.

Together, upregulation of *Kirrel1* during YAP-driven liver regeneration and restriction of YAP activity and the associated regenerative response suggest that KIRREL1 acts as a negative feedback regulator of YAP activity during DR in mice.

## Discussion

The Hippo pathway functions as the main signaling hub where various mechanisms sensing cell–cell contact, cell density, cell polarity, cell shape, mechanotransduction, and tissue architecture

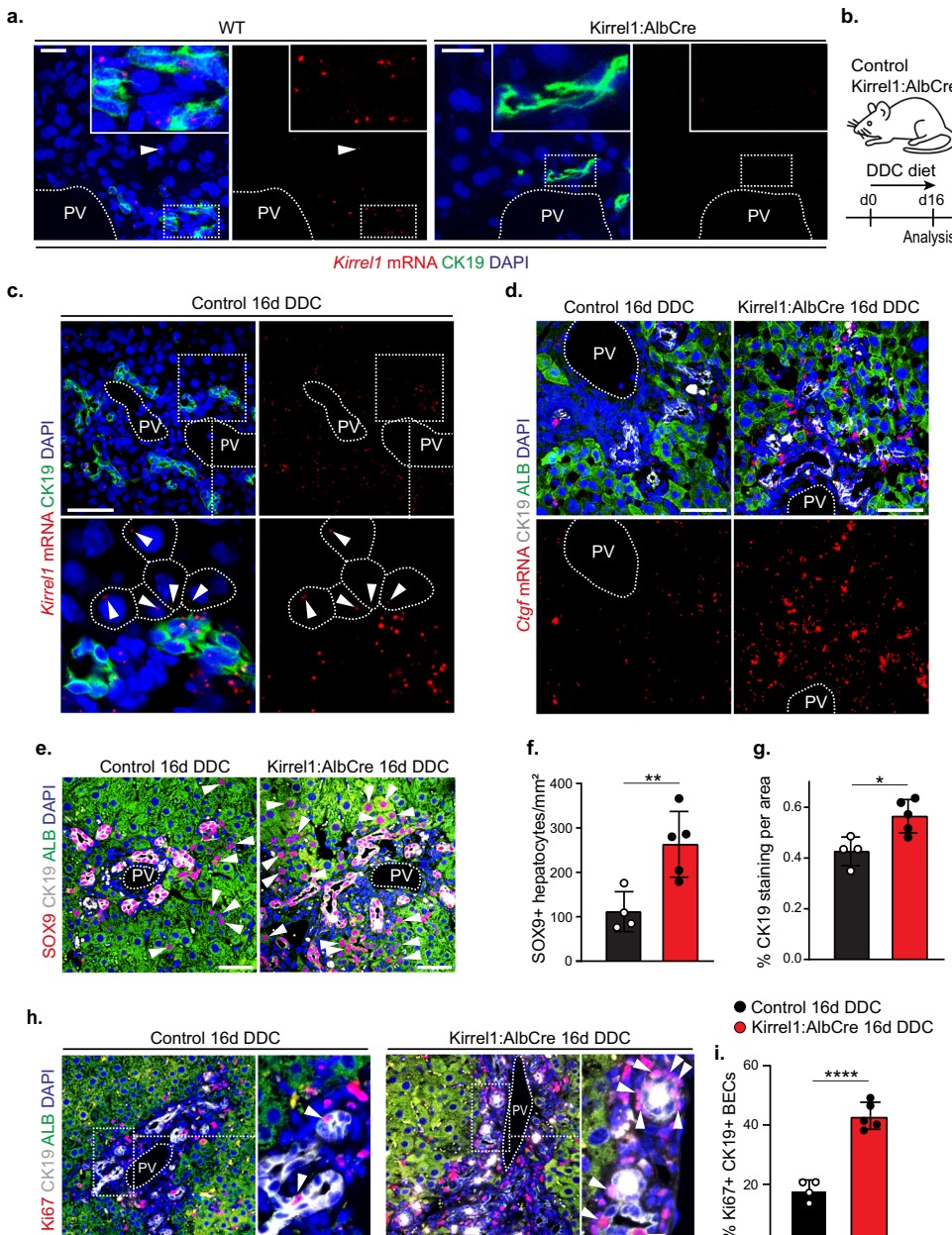

**Fig. 6 KIRREL1 restricts YAP/TAZ activity during ductular reaction (DR) in mice. a** *Kirrel1* ISH co-staining with the BEC marker CK19 in Kirrel1:AlbCre and WT control mice. PV portal vein. Arrowheads indicate *Kirrel1* ISH signals in periportal hepatocytes. Scale bar, 50 μm. **b** Scheme depicting 16 days DDC diet injury in Kirrel1:AlbCre and control mice. **c** *Kirrel1* ISH co-staining with CK19 in control mice 16 days post DDC diet. PV, portal vein. Arrowheads indicate *Kirrel1* ISH signals in encircled hepatocytes. Scale bar, 100 μm. **d** ISH for the YAP target gene *Ctgf* co-stained with CK19 and the hepatocyte marker Albumin (Alb), showing increased YAP signaling during DDC-induced DR in Kirrel1:AlbCre mice. PV, portal vein. Scale bar, 100 μm. **e–g** Co-staining for CK19, ALB, and SOX9 (**e**) in the indicated mice (arrowheads mark SOX9 + hepatocytes). Quantification indicates increased numbers of SOX9 + hepatocytes (**f**) and the percentage of CK19 staining per area (**g**, normalized to number of portal veins in the analysis area) in Kirrel1:AlbCre compared to control mice following 16 days DDC diet (IHC stainings used for quantification are provided in Fig. S6e). Data represent mean ± SD. Two-tailed unpaired *t*-test was used with groups of *n* = 4 control and *n* = 5 Kirrel1:AlbCre mice (6f, g). *p* values are 0.0090 (**, **f**), 0.0125 (*, **g**). Scale bar (**e**), 100 μm. **h, i** Co-staining for Ki67, CK19, and ALB (**h**) and quantification (**i**) indicates increased number of proliferating Ki67 + CK19 + BECs (arrowheads) in Kirrel1:AlbCre compared to control mice. PV portal vein, Data represent mean ± SD. Two-tailed unpaired *t*-test was used with groups of *n* = 4 control and *n* = 5 Kirrel1:AlbCre mice. *p* value is 0.00009 (****, **i**). Scale bar, 100 μm (**h**).

are integrated to regulate cell growth[39]. The discovery of NF2/Merlin and Expanded provided the first indication of how junctional proteins play critical roles in the Hippo pathway[29]. Since then, several tight and adherent junction proteins, as well as cell polarity complex proteins have been discovered as upstream regulators of the Hippo pathway[40–46]. The Hippo pathway was first identified in *Drosophila* and since then the fruit fly has served

as the discovery engine for the field. On the other hand, although the core components are highly conserved between *Drosophila* and mammals, upstream regulators of the Hippo pathway are less conserved. Systemic loss-of-function screens of mammalian Hippo pathway to identify upstream regulators are limited. Well-based siRNA screening is not ideal for Hippo signaling; as Hippo signaling is regulated by cell density, cells with decreased

proliferation would have increased YAP/TAZ activity, which significantly increases the noise of the screen. Here, we overcome this issue by using a pool-based CRISPR screen and identify KIRREL1 as an important upstream regulator of mammalian Hippo signaling. Since the Hippo pathway is influenced by a multitude of intrinsic and extrinsic signals, similar screening strategy, as depicted in this study, can be used to identify additional upstream regulators of mammalian Hippo signaling in response to different stimuli.

KIRREL1 was first identified in a gene trapping screening and named as NEPH1 due to its structural similarity to NEPHRIN, a protein associated with congenital nephrotic syndrome[32]. The NEPHRIN-NEPH1-Podocin trimeric protein complex forms the slit diaphragm (SD) between interdigitating podocytes and functions as a critical size and charge filtration barrier to prevent proteinuria[23,24,47]. Mice deficient in *Kirrel1* develops severe proteinuria due to the effacement of podocyte foot processes[32]. Loss of *Drosophila* orthologues of *KIRREL1-dumbfounded* (*duf*, also known as *kirre*) and *roughest* (*rst*) disrupts the nephrocyte diaphragm[48], indicating the evolutionarily conserved function of KIRREL1. Importantly, unlike KIRREL2, KIRREL3, and NEPHRIN, which are predominantly expressed in kidney podocytes[47], KIRREL1 is broadly expressed in many tissues (Fig. S5e). However, the function of KIRREL1 beyond filtration barrier in the kidney remains poorly understood. Our study thus expands KIRREL1's biological function outside of the kidney and demonstrates KIRREL1-mediated regulation of Hippo signaling.

Although the Hippo pathway was initially characterized as a linear pathway, recent studies suggest that SAV1 and NF2 function in parallel targeting MST1/2 and LATS1/2, respectively, to the plasma membrane to fully activate the kinase cassette[15]. Therefore, mechanisms that control membrane targeting of SAV1 are expected to have strong impacts on the activation of the Hippo signaling. Our data suggest that KIRREL1 regulates the Hippo pathway through recruitment of SAV1 to cell–cell contacts. Since KIRREL1 mediates cell-cell adhesion through its extracellular domain, cell–cell interaction might regulate KIRREL1-SAV1 interaction and the function of KIRREL1 in Hippo signaling. The extracellular domain (ECD) of KIRREL1 harbors five extracellular immunoglobulin (Ig) domains of the C2 type and has been shown to form trans-homodimeric interactions between adjacent cells[23,24,35,49]. Consistent with this hypothesis that KIRREL1-mediated cell adhesion suppresses YAP activity, knockout of *KIRREL1* increases YAP activity in neighboring cells (Fig. 4d). Homophilic interaction of KIRREL1 ECDs between adjacent cells, therefore, might serve as a "sensing" mechanism for cell–cell interaction resulting in its ICD-mediated recruitment of SAV1 to the cell–cell contact sites to activate the Hippo kinase cascade (Fig. S8).

In *Drosophila*, Echinoid (Ed) was identified as an upstream modulator of the Hippo pathway that facilitates Sav recruitment to the plasma membrane[50], and whether Echinoid has a homolog in mammals has not been clear. Echinoid and KIRREL1 have similar domain structures; Echinoid has seven Ig-like domains followed with one Fibronectin type-III domain in its extracellular domain while KIRREL1 has five Ig-like domains in its extracellular domain. Echinoid and KIRREL1 also have similar functions; both Echinoid and KIRREL1 bind to SAV through the C-terminus of their intracellular domain. Therefore, KIRREL1 appears to be a functional homolog of Echinoid.

Inhibition of Hippo signaling has been reported in many types of tumors and correlates with poor prognosis[6,51,52]. Although knockout of *NF2* is expected to strongly increase YAP/TAZ activity in most cell lines, only a fraction of cell lines show enhanced proliferation upon *NF2* knockout in pan-cancer CRISPR proliferation screens (Fig. S5a), suggesting that not all

cancer lines are sensitive to YAP activation to the same degree. Strikingly, correlation analysis identifies KIRREL1 as the cell surface protein showing the strongest correlation with known regulators of Hippo signaling (Fig. 5a–c). These results suggest that KIRREL1 likely serves as an important upstream sensor of the Hippo pathway during cell proliferation in tissue culture. Cell lines sensitive to *KIRREL1* KO also express high levels of *KIRREL1* (Fig. 5d). Since KIRREL1 is a target gene of YAP/TAZ, these cell lines might have high basal YAP/TAZ activity, and *KIRREL1* KO further enhances YAP/TAZ activity and increases proliferation in these cell lines. Additional studies are needed to further examine the function of KIRREL1 in different cancers.

Accumulating evidence suggest that YAP/TAZ activity is critical for tissue repair upon injury[8,52]. We and others demonstrated the critical role of a YAP/TAZ-dependent transcriptional program during BEC activation and expansion during liver regeneration in a DR-inducing liver damage model, as well as during reprogramming of periportal hepatocytes[36–38]. Our finding that hepatic *Kirrel1* expression correlates with YAP/TAZ activity in naive mouse livers and DR in mice and human, supports that *Kirrel1* expression is regulated by YAP/TAZ. Importantly, increased YAP/TAZ activity and associated regenerative responses substantiate the role of Kirrel1 as a negative feedback regulator. Increased hepatocyte reprogramming, BEC proliferation and resulting DR in Kirrel1:AlbCre mice suggest that Kirrel1 limits YAP/TAZ activation in response to DDC-induced liver injury. Interestingly, unlike in *Sav1* KO mice[53], we did not observe hepatomegaly in Kirrel1:AlbCre mice, suggesting the existence of redundant mechanisms to promote SAV1 membrane targeting in hepatocytes. Consistent with this, functional redundant modules have been described in the Hippo pathway; MAP4K4/6/7 and MST1/2 kinases act in parallel to activate LATS1/2 in mammals[54,55]. In the DDC injury model, YAP activity is markedly increased in BECs and periportal hepatocytes, which leads to further increase of Kirrel1 expression in these cells. This is likely the reason that Kirrel1 plays a dominant role over other mechanisms in the DDC model. We did not observe altered liver physiology in adult mice with *Kirrel1* deletion, suggesting normal postnatal liver development. However, AlbCre-mediated deletion occurs rather late during embryonic liver development and is incomplete at birth[56]. Future studies using different Cre lines will be necessary to assess the role of Kirrel1 during liver development.

KIRREL1 is broadly expressed in different tissues and knockout of KIRREL1 increases proliferation of tumor cells of different lineages. Whether KIRREL1 expression is correlated with YAP/TAZ activity in other tissues requires further investigation. Future studies with tissue-specific knockout of *Kirrel1* in other tissues will deepen our understanding of functions of KIRREL1 in YAP/TAZ-mediated tissue homeostasis and injury repair.

## Methods

**Ethics and animal welfare**. Glass slides with Formalin-fixed paraffin-embedded (FFPE) sections from 4 patient livers (1 female, age 81; 3 males, age 41, 64, and 86) were obtained from the University Hospital Basel Tissue Bank. The biopsies were originally acquired for routine diagnostic and patients signed a general informed consent for the use of remaining tissue for research purposes in accordance with the Swiss Federal Human Research Act (HRA). Patients did not receive compensation. The study was approved by the ethics committee of Northwest and Central Switzerland (EKNZ) as part of the EKNZ (former EKBB:361/12). All animal experimentation was conducted in accordance with animal law and approved by the cantonal veterinary office Basel-Stadt, Switzerland.

**Cell culture**. HEK293A (R70507, Thermo Fisher) and HEK293T (CRL-11268, ATCC) cells were grown in DMEM (Gibco) supplemented with 10% FBS and penicillin (100 units/ml)-streptomycin (100 μg/ml) (Gibco). GAMG, DAOY, NCI-H2030, and IGR-39 cell lines were originated from the CCLE[57], banked at Novartis Cell Bank, and authenticated by STR profiling. GAMG cells were cultured in

DMEM supplemented with 10% FBS and penicillin–streptomycin. DAOY and NCI-H2030, cells were maintained in RPMI medium 1640 (Gibco) supplemented with 10% FBS and penicillin–streptomycin. IGR-39 cells were cultured in DMEM media supplemented with 15% FBS and penicillin–streptomycin. All cell lines were maintained at 37 °C in a 5% $CO_2$ incubator.

**Cell line generation, plasmids, and transfection.** To generate HEK293A *KIRREL1* or *SAV1* knockout cells, gRNAs were cloned into lentiviral vector pNGx-LV-g003 that expresses gRNA under U6 promoter and RFP-T2A-Puro under UbiC promoter. HEK293A cells stably expressing Cas9 were transduced with lentivirus expressing *KIRREL1* or *SAV1* gRNAs and selected by Puromycin. To generate *KIRREL1* knockout cancer cells, gRNAs were cloned into 2-in-1 lentiviral vector pNGx-LV-gc006 that expresses gRNA under U6 promoter and codon Cas9-T2A-Puro under EF1α promoter. Cancer cell lines were transduced with lentivirus expressing Cas9 and *KIRREL1* gRNA and selected with Puromycin. To avoid clonal variation, pool of knockout cells were used for all experiments. For CRISPR knockout, following gRNAs were used for transduction followed by selection with antibiotics—

 Control (sg*AAVS*)—5′ GGGGCCACTAGGGACAGGAT
 *KIRREL1* gRNA1—5′ GAGTGAGGATCCAGACGAGG
 *KIRREL1* gRNA2—5′ GGAGCCAGCTGACCAGACGG
 *SAV1* gRNA—5′ GAGCGAGAAGGACTTCCTC
 *YAP/TAZ* gRNA—5′ AATGTGGATGAGATGGATAC

For shRNA-mediated depletion of KIRREL1, following shRNAs were cloned into pLKO-Tet-On vector.

 Control shRNA—5′ CAACAAGATGAAGAGCACCAA
 *KIRREL1* shRNA1—5′ GGCCATCTACTCGTCGTTTAA
 *KIRREL1* shRNA2—5′ AACCTCACAAGACACAGGC

For overexpression studies, KIRREL1 cDNA was cloned into pCDH-MCS-T2A-Puro (System Bioscience). SAV1 cDNA was cloned into EGFP-N1 (Clontech). Deletion mutants were generated using Q5 Site-Directed Mutagenesis Kit (New England Biolabs).

Transfection was done by using Fugene 6 (Promega) or Lipofectamine 3000 (Invitrogen) according to the manufacturer's instructions. For stable cell line generation, lentiviral vectors were packaged into HEK293T cells using standard virus packaging protocol and transduced into indicated cells followed by selection with appropriate antibiotic.

Sequence of gRNA, shRNA, and oligos for generating cDNA expression constructs can be found in Supplementary Table.

**Genome-wide CRISPR screen and data analysis.** We designed five gRNAs for each gene using Illumina Human BodyMap 2.0 and NCBI CCDS data sets. The gRNA library containing 90,000 gRNAs was synthesized using array synthesis and cloned into lentiviral vector pNGx-LV-g003 that expresses gRNA under U6 promoter and RFP-T2A-Puro under UbiC promoter. HEK293A GTIIC-GFP$^{PEST}$ Cas9 cells were seeded into two five-chamber cell stacks (Corning) at $67 \times 10^6$ cells/stack. Twenty-four hours after plating, cells were infected with gRNA lentivirus library at an MOI 0.5. Twenty-four hours following transduction, the culture medium was replaced with fresh medium containing 2 μg/ml Puromycin. Cells were maintained in medium containing Puromycin and split twice. During the second split, cells were seeded into five cell stacks at a density of $80 \times 10^6$ cells/stack and grown for six additional days without further passage to reach high density. Fourteen days post Puromycin selection, cells were harvested and sorted using BD FACS Aria Cell Sorter. Cells were at high density at the time of cell harvesting for FACS sorting to increase the chance of identifying positive regulators of Hippo signaling. Top 20% GFP-high and top 20% GFP-low populations of RFP positive cells were collected by FACS sorting. Genomic DNA was collected using the QIAamp DNA Blood Maxi Kit (Qiagen) following manufacturer instructions and subjected to Illumina DNA sequencing for barcode counts. Raw sequencing reads were aligned to the appropriate library using Bowtie, allowing for no mismatches, and counts were generated. The R software package DESeq2 was used to evaluate differential gRNA representation in the form of log2 fold change and *p*-value between the GFP-high and the GFP-low samples. A robust *z*-score for each gRNA was calculated using the median and mean-absolute deviation across the log2 fold changes. To summarize the results at the gene level, the gRNAs are ranked by the robust *z*-score, and the statistical significances for each gene enriched toward higher rank (RSA up) were evaluated using the Redundant siRNA Activity (RSA) algorithm. The RSA *p*-value is shown along with the log2 fold change of the five gRNAs per gene for the indicated set of genes.

**Flow cytometry assay.** HEK293A GTIIC-GFP$^{PEST}$ Cas9 cells expressing indicated gRNAs mixed with (or without) parental cells were plated at $0.5 \times 10^6$ cells/well in a 12-well plate. Cells were trypsinized 48 h after plating, resuspended in cell culture medium and then subjected to flow cytometry analysis using a Cytoflex flow cytometer (Beckman Coulter Life Sciences). Cells were first gated on size (FSC-A) and granularity (SSC-A) to exclude cellular debris and sequentially gated using FSC-H and FSC-A to eliminate doublets. The single cell populations were then further gated for the subset of RFP positive or RFP negative (as indicated). The GTIIC-GFP$^{PEST}$ reporter activity in these cell population was measured by using the FITC

channel, and raw data was analyzed by FlowJo software. Gating strategy is shown in Fig. S9.

For KIRREL1 membrane staining, cells were collected using cell dissociation buffer (Gibco) and resuspended in FACS buffer (PBS, pH 7.2, with 1% BSA). After blocking, cells were incubated with anti-Myc-Alexa fluor 488 (Cell Signaling Technology) antibody for 1 h at 4 °C. After washing with FACS buffer for three times, cells were stained with SYTOX Blue (Thermo Fisher Scientific) and subjected to FACS analysis using CytoFLEX flow cytometer (Beckman Coulter) and analyzed with FlowJo software.

**RNA extraction, reverse transcription, and quantitative RT-PCR.** Cell were plated at $0.5 \times 10^6$ cells/well in 12-well plate. Forty-eight hours post cell seeding, total RNA was extracted by using RNeasy Plus mini kit (QIAGEN) and reverse transcribed with TaqMan reverse transcription reagents (Life Technologies) according to the manufacturer's instructions. Gene expression levels were detected by TaqMan probes, and all the experiments were performed in quadruplicates in QuantStudio 7 Flex Real-Time PCR System (Applied Biosystems). The comparative cycle threshold value (Ct) was determined for each transcript and normalized against the housekeeping gene (GUSB or GAPDH). The relative expression of each mRNA was determined using the comparative ΔΔCt method. Taqman probes used in this study were obtained from Thermo Fisher Scientific and are as below—*CTGF* (Hs00170014_m1), *ANKRD1* (Hs00923599_m1), *CYR61* (Hs00998500_g1), *LATS2* (Hs00324396), *NF2* (Hs00184311), *KIRREL1* (Hs00217307), *GUSB* (Hs00939627_m1), and *GAPDH* (Hs02758991_g1).

**Cell lysis, immunoblotting, and immunoprecipitation.** For immunoblotting, cells were lysed using RIPA buffer (25 mM Tris-HCl pH 7.6, 150 mM NaCl, 1% NP-40, 1% sodium deoxycholate, 0.1% SDS) (Thermo Fisher Scientific) supplemented with Protease Inhibitor Cocktail and Phosphatase Inhibitor followed by centrifugation at $15,000 \times g$ for 10 min at 4 °C. Protein concentration was measured using the DC Protein Assay Kit (Bio-Rad) and equal amount of proteins were analyzed by SDS-PAGE and western blot analysis using standard procedure.

For co-immunoprecipitation experiments, cells were lysed using NP40 Lysis buffer (Boston BioProducts). Equal amounts of cleared cell lysates were adjusted to the same volume and incubated with anti-Flag tag (Sigma Aldrich) or anti-Myc tag (Thermo Fisher Scientific) magnetic beads for overnight incubation at 4 °C with gentle agitation. Beads were washed four times before elution with SDS sample buffer for immunoblot analysis.

Antibodies used in this study are anti-Flag tag (Cat# F1804, Sigma Aldrich), anti-Myc tag (Cat# 2272S, CST), anti-HA tag (Cat# 3724P, CST), anti-V5 (Cat# 13202S, CST), anti-GAPDH (Cat# 8884S, CST), anti-SAV1 (Cat# 3507S, CST), anti-GFP (Cat# A-11122, Thermo Fisher), anti-Lamin A/C (Cat# 2032, CST), anti-YAP (Cat#14074, CST, anti-YAP/TAZ (Cat#8418, CST), anti p-YAP (Cat# 4911, CST) anti Histone H3 (Cat# 4499, CST). See detailed information in Supplementary Table 3.

**Protein purification and Ni-NTA pull-down assay.** The N-terminal and the C-terminal KIRREL1 ICD (ICD77 C-ter or ICD80 N-ter) were cloned into Champion™ pET SUMO Expression System (Thermo Fisher Scientific) and transformed into BL21 (DE3) strain (Stratagene). Bacterial cultures were grown at 37 °C and induced with isopropyl 1-thio-β-D-galactopyranoside and incubated at 30 °C overnight. Proteins were purified by a Ni-NTA affinity column (GE Healthcare). For Ni-NTA pull-down assay, equal amount of purified His-SUMO tagged KIRREL1 ICD fragments (ICD$_{77}$ C-ter or ICD$_{80}$ N-ter) were bound to Ni-NTA resin (QIAGEN), and incubated with HEK293T cell lysates expressing MST1 or SAV1 in NP40 lysis buffer overnight at 4 °C with gentle agitation. The beads were washed four times before elution with SDS sample buffer for western blot analysis and coomassie staining. For KIRREL1 ICD and SAV1 direct binding, purified His-SUMO tagged KIRREL1 ICD fragments bound to Ni-NTA resin were incubated with purified SAV1 (Origene) in NP40 lysis buffer for 4 h at 4°C. The beads were washed four times and bound proteins were resolved on SDS-PAGE and examined by western blot analysis.

**Nuclear fractionation.** HEK293A cells were plated in 10 cm dishes at high or medium density as indicated. Cells were washed with PBS and cell pellets were resuspended in hypotonic buffer (10 mM Tris-HCl pH 7.5 and 10 mM KCl) supplemented with Protease Inhibitor Cocktail and Phosphatase Inhibitor followed by freeze-thaw for four times. The lysates were centrifuged at full speed for 10 min. The pellet was washed with hypotonic buffer and resuspended in RIPA buffer containing Protease Inhibitor Cocktail and Phosphatase Inhibitor and sonicated using a Diagenode Bioruptor 300 (30 s on, 30 s off, 5 cycles), followed by centrifugation at $15,000 \times g$, 4 °C for 10 min. The supernatant was collected, and protein concentration was measured using the DC Protein Assay Kit (Bio-Rad).

**Luciferase reporter assay.** HEK293A cells were co-transfected with YAP and vector or plasmid encoding wild-type and mutant forms of KIRREL1 together with GTIIC-luciferase plasmid and *Renilla* luciferase pRL-TK plasmid (Promega) using Lipofectamine 3000 (Thermo Fischer Scientific) in 96-well plates. After 48 h, cells were lysed, and luciferase activity was measured using Dual-Glo® Luciferase Assay

System (Promega) using EnVision Multimode Plate Reader (PerkinElmer). The experiment was performed three times in quadruplicates, and data from representative experiments were shown. The GTIIC firefly luciferase activity was normalized to *Renilla* luciferase activity in each well. The figures show the mean ± standard deviation.

**Immunofluorescence staining**. Cells were plated on glass coverslips in 6-well plates and fixed with 2% formaldehyde solution for 10 min at room temperature. Following washing with PBS three times, cells were permeabilized with 0.5% Triton X-100 solution (25 mM HEPES pH 7.4, 50 mM NaCl, 1 mM EDTA, 3 mM MgCl$_2$, 300 mM Sucrose, 0.5% triton-X) at 4 °C for 5 min and incubated with indicated primary antibodies at 37 °C for 1 h. Cells were then incubated with secondary antibodies conjugated with Alexa 488 or Alexa 555 (Invitrogen) for 1 hr at 1:1000 dilution and washed for five times. Images were captured using a Zeiss Airyscan confocal microscope.

**Proliferation assay**. Indicated cancer cell lines were seeded at $0.15 \times 10^6$ cells/well to $0.25 \times 10^6$ cells/well depending on the cell size in four 12-well plates in triplicates in appropriate media supplemented with FBS and penicillin (100 units/ml)-streptomycin (100 μg/ml) (Gibco). Cell numbers were counted on day 2, 4, 6, and 8 after seeding by trypan blue exclusion on a ViCELL instrument (Beckman Coulter). Data represent mean ± SD.

**Mouse models and in vivo mouse studies**. We generated transgenic mice with LoxP-flanked *Kirrel1* exon 5 (floxed *Kirrel1* mice) and crossed these mice with Albumin-Cre (AlbCre) mice[58], enabling Cre-mediated gene deletion of *Kirrel1* in both hepatocytes and biliary epithelial cells[37]. Mice were generated in Cambridge, MA and animal experimentation was performed in Basel, Switzerland. C57Bl/6 WT mice or littermate controls of Kirrel1:AlbCre mice (floxed Kirrel1 mice without AlbCre) were used as controls as indicated in the Figures and legends. Liver damage and resulting ductular reaction was induced with diet supplemented with 0.1 % 3,5-Dicarbethoxy-1,4-dihydrocollidine (DDC; Cat#137030, Sigma-Aldrich) for 16 days. Interruption of the diet for 48 h allowed recovering body weight, while all mice within an experiment (mutants and controls) had similar DDC diet regimen. Body weight and wellbeing of the animals were monitored daily until the end of the experiment. Mice had C57Bl/6 background and we used 16–21-week-old males for the experiments ($n = 4$ control and $n = 5$ Kirrel1:AlbCre mice with DDC injury; $n = 3$ WT and $n = 2$ Kirrel1:AlbCre naïve mice). All mice had unrestricted access to water and food. Our animal facilities comprise an SPF animal breeding facility and a clean facility for experimental surgery and physiology. Biosecurity and pathogen exclusions follow the Federation of European Laboratory Animal Science Associations (FELASA) health monitoring guidelines, and animals are screened quarterly. Mice were housed in Allentown XJ individually ventilated cages in a 12:12 light:dark cycle. Environmental enrichments included nestlets, wood sticks and mouse houses.

**Immunohistochemistry (IHC) and in situ hybridization (ISH)**. IHC was performed using either a Roche Discovery XT or manually as described[59]. Primary antibodies used in this study were Rabbit anti-Ki67 (Cat# RM-9106, Thermo Scientific), Rat anti-CK19 (DSHB, TROMA-III), Rabbit anti-SOX9 (Cat# AB5535, Millipore), Rabbit anti-Glutamine Synthetase (Cat# AB49873, Abcam), Chicken anti-Albumin (Cat# SAB3500217, Sigma-Aldrich), Rabbit anti-Iba1 (Cat# AB178846, Abcam). The TROMA-III, developed by R. Kemler, was obtained from the Developmental Studies Hybridoma Bank developed under the auspices of the NICHD and maintained at The University of Iowa, Department of Biology, Iowa City, IA 52242. HRP- (Nichirei Biosciences Inc.), DyLight- or Cy-coupled (Jackson ImmunoResearch) secondary antibodies were used for primary antibody detection, and either Mayer's Haematoxylin (Dako, S3309) or DAPI (Sigma-Aldrich, D9542) were used for counterstaining. For representative images, whole-liver lobes were examined histologically in multiple replicates. CK19 and SOX9, Ki67, HNF4α and GS quantification was performed on DAB-stained formalin—fixed paraffin-embedded (FFPE) sections. The quantification of Ck19 and Iba1 was performed using the Color deconvolution algorithm in the imagescope software. For Ck19 the total positive signal was normalized to the number of portal veins per analysis area.

Sox9 quantification was done in the Imagescope software using a nuclear count algorithm, The number of Sox9 positive hepatocytes was normalized to the analysis area.

ISH was performed as described[60] for *Ctgf* (probe VB1-15089-VT, Affymetrix) and *Kirrel1* (probe VB6-3214990-VT, Life Technologies). ISH images were acquired using an Olympus laser-scanning confocal microscope FV3000. Signal intensity was adjusted on each channel according to their histograms and using fixed parameters across the whole batch of pictures. Co-staining of the ISH with CK19 and Albumin antibodies allowed for localization of the respective ISH signal.

For ISH on human livers, we obtained formalin-fixed paraffin-embedded (FFPE) sections of human needle liver biopsy tissue from the University Hospital (see Ethics and animal welfare section for details). Samples from four patients (one female, age 81; three males, age 41, 64, and 86) with liver injury-associated ductular reaction were analyzed. ISH was performed using the RNAscope® Multiplex

Fluorescent Reagent Kit v2 (ACD Bio, Europe SRL, Cat. No. 323100) according to the manufacturer's instructions. Briefly, 5 μm liver FFPE sections were deparaffinized, blocked for 10 min with hydrogen peroxide, followed by target retrieval for 30 min and air-dried overnight. Sections were pre-treated with protease solution for 30 min at 40 °C and afterwards hybridized for 2 h at 40 °C with *KLF6* (#489301) and *KIRREL1* (#1063771) probes. Sections were subsequently amplified according to the manufacturer's instructions and stained for detection with Opal™ fluorophores 570 (Akoya Bioscience, FP1488001KT) and 690 (Akoya Bioscience, FP1497001KT). Co-staining of the ISH samples with CK19 antibody (rat-anti CK19 (DSHB, TROMA-III)) was performed to allow for the localization of ductular reaction areas and the respective ISH signal. At the end, samples were counterstained with DAPI and mounted. ISH images were acquired using a laser scanning confocal microscope (Olympus FV3000) and fixed parameters. From each sample, ten different fields of view (FOV) of ductular reaction areas, as well as liver parenchyma were captured. Images were quantified using ImageJ, where the signal intensity was adjusted on each channel according to their histograms and after signal segmentation, the percentage (%) of the total area for each signal was determined.

**Analysis of serum markers**. Serum was collected during dissection of Kirrel1:AlbCre and control mice following 16 days DDC diet. Serum markers from Kirrel1:AlbCre and control mice following were analyzed using SPOTCHEM II LIVER-1 (Cat# 77182, Arkray) and SPOTCHEM II ALP (Cat# 77176, Arkray).

**Statistics and reproducibility**. Experiments were not randomized and repeated at least three times unless indicated differently. Data are presented as the mean ± SD. For all graphs, data are presented relative to their respective controls. In vivo experiments were performed once but including several age-matched and sex-matched mice per genotype and condition (as indicated in the Figure legends). Stainings were established and then used to generated the provided images (Fig. 6a, c, d, e, h and S6a, c, d) exemplary for the mice from the respective in vivo experiments ($n = 4$ control, $n = 5$ Kirrel1:AlbCre mice with DDC injury; $n = 3$ WT, $n = 2$ Kirrel1:AlbCre naïve mice). IBA1 staining and quantification (Fig. S7c, d) were performed in livers from $n = 3$ control and $n = 4$ Kirrel1:AlbCre mice. The quantifications shown in Fig. 6f, g are based on the staining in Fig. S6e ($n = 4$ control, $n = 5$ Kirrel1:AlbCre mice with DDC injury). Staining for human liver biopsies was established and validated on control human liver tissue and performed once on the liver biopsies from four patients shown in Fig. S7e. All statistical analyses were performed using GraphPad Prism 8 (Graphpad software Inc). Significance was determined by using unpaired two-tailed *t*-test or two-way ANOVA, depending on the number of samples that were analyzed per experiment (two or more, respectively). $p < 0.05$ was considered to be statistically significant. Details on statistical tests and significance of differences for each experiment were provided in the respective Figure legends.

**Reporting summary**. Further information on research design is available in the Nature Research Reporting Summary linked to this article.

## Data availability

Correspondence and request for any materials used in this study should be sent to feng.cong@novartis.com. All relevant data supporting the findings of this study are available within the paper and Supplementary information files. Raw data and uncropped gel images of all figures are included in the Source Data file. We used the following publicly available datasets: the DepMap portal for dependency scores (https://depmap.org/portal) and GTEx database for tissue-specific gene expression (https://gtexportal.org/home/gene). All other data supporting the findings of this study are available from the corresponding author on reasonable request. Source data are provided with this paper.

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

## Acknowledgements

We thank Akos Szilvasi, Alan Ho, Melissa Paziuk, Chad Vickers for technical assistance. We thank Yan Feng and David Glass for comments and advice.

## Author contributions

A.P., J.S.T., and F.C. conceived and designed the study. B.L. performed the CRISPR screen. J.R.-H. generated CRISPR library. A.L. and C.R. performed the Next generation sequencing. Z.Y.L. and F.S. analyzed the CRISPR screen data. B.L., O.C., L.J., and R.Z. performed initial validation of the CRISPR screen data. A.P. carried out in vitro experiments. H.L., T.B.N. generated Kirrel1 floxed mice. J.S.T. designed the mouse study and S.A., T.S., O.E., L.P.-P., V.O., and J.S.T. performed and analyzed in vivo mouse experiments. L.M.T. evaluated and provided human liver samples. A.P., J.S.T., and F.C. wrote the paper. All authors read and approved the final manuscript.

## Competing interests

The authors declare no competing interests.
