## [Peer Review File · Nature Communications]

Cell adhesion molecule KIRREL1 is a feedback regulator of Hippo signaling recruiting SAV1 to cell-cell contact sitesReviewers' Comments:

Reviewer #1:

Remarks to the Author:

In this manuscript, Atanu et al identify KIRREL1 as an upstream regulator of Hippo signaling that binds to SAV1. The authors identify the interaction domains in KIRREL1 and SAV1 and show that KIRREL1 expression is induced upon YAP activation and acts as negative feedback for Hippo signaling. The manuscript is clearly written, and the data presented underscores most of the claims and provides new insight for the YAP research community. The *in vivo* data presented that underlines the relevance of their findings is not so convincing. The cancer cell lines show indeed a strong correlation of KIRREL1 with known Hippo signaling. However, authors could provide evidence on human liver biopsies that shows that there is a correlation between KIRREL1 and YAP downstream targets. In livers of alcoholic hepatitis for instance in which a strong DR takes place, one would expect high KIRREL1 levels in the expanding BECs.

Please find below some additional comments that could further improve the manuscript.

Figure 1.

The screen could be explained better in the M&M; "HEK293A GTIIC-GFPPEST Cas9 cells were transduced with gRNA lentiviral library at MOI 0.5. Fourteen days post lentiviral transduction, cells were grown to high cell density, and sorted using BD FACS Aria Cell Sorter". Were cells passaged in between, were they selected?

The raw FACS data used to sort the GFP-High and Low population should be shown? Not just a schematic representation. How was the sort done, on FSC or SSC vs GFP? Were there clear populations visible? This diagram does not give us much information..

Since high and low GFP populations are sorted, the graph in Fig1b should also show the gRNAs overrepresented in the GFP-low populations for instance (or the top 10 could be given in a table in sup). These enhanced gRNAs should be enhancers of Hippo signaling. It would strengthen the paper to show that such a LOF screen could also identify known enhancers of Hippo.

KIRREL1 gRNA is used in 1C, but the efficiency is not shown, while this is done for the shRNA of KIRREL (in Figure S4).

"Control and KIRREL1 knockout cells were plated at high or medium density and subjected to western blot analysis." What does this mean? Are cells plated at the same density and then grown until medium or high density or immediately seeded at high or medium density? Since the low, medium and high density culture is used several times, images could be shown to illustrate the density. In 1e, it would be good to have a low density control blot next to these medium/high density blots and to show YAP as well. Is KIRREL only affecting TAZ in these conditions?

For f and g, it is not clear at what density the cells were taken for FACS or target gene determination, please specify because density does matter for all the experiments.

Figure 2.

Clearly shows that overexpression of KIRREL affects the expression of TAZ, and decreases target expression of YAP/TAZ downstream genes. Authors do show that overexpressed TAZ-V5 is affected. It would be more convincing to show that endogenous YAP and TAZ is affected by overexpressed KIRREL. This would be more in line with the target gene analysis and the luciferase assays shown in the rest of the figure, in which no TAZ is overexpressed.

The quadruplicate experiments for luciferase, are they biological repeats or technical repeats of one transfection experiment? The M&M states "the experiment was performed in quadruplicates" which seems to suggest that the experiment was done only once? In the figures it would be good to mention "GTIIC-luciferase" (or anything else indicating which luciferase it is) above the luciferase graphs.

Figure 3

Clear data showing that the c-term region of the ICD of KIRREL is necessary for the binding with the N-term of SAV-1 and that deletion of this ICD domain affects the luciferase assay. Does it also affect the gene expression of YAP/TAZ downstream genes in a similar way? How about stability of the

endogenous YAP/TAZ proteins when this ICD3 mutant is overexpressed?

Figure 4,

The FACS plots show that there seems to be more cells with a higher GFP signal, but there is no clear shift of low to high expressing cells. Can the authors explain why this is.

Figure 4S shows that the levels of both KIRREL and SAV are very low, still this does not seem to affect all cells as evidenced by the plots? Is this due to density in the dishes? Authors could discuss this minor affect of knocking down KIRREL and SAV on the transgene.

In 4b. authors claim that there is no significant difference between single and double knock-down, but significance is not given. One could indicate NS between the single and double knock-down bars (if this is indeed the case).

In 4e one shows that KIRREL expression is correlated with density. It would be nice to know what density we are talking about (images of the cells in culture) and to see mRNA levels of targets of YAP (ANKRD1, CTGF etc) alongside KIRREL1 expression. Is the expression really correlated or are there waves of KIRREL1 and YAP-targets?

Figure 5: "For generating KIRREL1 knockout cancer lines, cancer cell lines were transduced with lentiviral vectors expressing both Cas9 and gRNA from a single vector and selected with Puromycin." Please indicate the vector used.

Figure 6.

This part shows that there is a greater DR when Kirrel1 is deleted by the AlbCre mouse. There is more YAP activation as evidenced by Ctgf ISHs and there is an increased number of CK19+ BECs and SOX9 positive periportal hepatocytes. Wouldn't KO of Kirrel in epithelial cells lead to a higher YAP activity during development, from the start of the AlbCre and thus affect liver growth or size? Authors do comment on this by suggesting the existence of redundant mechanisms to promote SAV1 membrane targeting in hepatocytes. But why are these mechanisms not in place in a the DDC injury model?

One aspect that is not shown is whether the injury induced by the 16days DDC in the AlbCRE Kirrel mice was different? If the absence of Kirrel in hepatocytes and BECs leads to more hepatocyte cell death due to this diet, than the DR would be greater as well. Authors should show ALT/AST, Billirubin etc. measurements and for instance liver to bodyweight ratios, showing that there is no difference in liver damage.

Minor comments:

Line 40: Zhao 2007 is used instead of a number

Line78: CRISPR screens or CRISPR screening iso screen

Line 252: (Fig. 5c) should be Fig. 5d.

Line:256 Fig. 5d should be Fig. 5e

Line:259: Fig. 5e should be Fig. 5f

Reviewer #2:

Remarks to the Author:

In an elegant approach combining genome-wide CRISPR screen with a reporter of YAP/TEAD activity, Paul et al. identified KIRREL1 as a novel upstream regulator of the Hippo signaling cascade. Using loss-of function and overexpression experiments, the authors convincingly show that KIRREL1 negatively regulates YAP/TAZ activity through binding to the adaptor protein SAV1 – an interaction that seems to be specifically relevant at cell-cell contact and might even influence YAP/TEAD activity in neighboring cells. Supporting evidence for the functional relevance of KIRREL1 in the control of Hippo/YAP signaling is provided by analysis publicly available screening data from different cancer cell lines and more importantly, from in vivo data in mice with liver-specific Kirrel1 deletion.

Overall, the manuscript is well written and the data presented support the authors' conclusions. With identification of a completely novel regulator of Hippo/YAP signaling, the results are of high relevance

in the field However, some parts of the manuscript outlined below should be addressed:

- 1) Figure 1e: only TAZ levels are shown. What happens to YAP expression and YAP/TAZ phosphorylation? If changes in YAP levels cannot be observed in HEK293 cells this should be addressed
- 2) Figure 4c: SAV1 and KIRREL also seem to co-localize the nuclear membrane. Is this observed on a regular basis? What could be the function at this nuclear interaction as it should not be relevant for sensing of cell-cell contacts?
- 3) Figure 4d: how is the difference in reporter GFP expression in RFP- vs RFP+ cells? This would help to show how important the influence of KIRREL1 is on neighboring cells in comparison the cell autonomous effects. Immunofluorescence staining for YAP would help to strengthen the results of this otherwise very elegant experiment.
- 4) Figure 5e: if possible, the authors should show YAP/TAZ protein levels or immunostaining for YAP in at least some of these cell lines
- 5) Figure 6: are there any changes in hepatic YAP localization by IHC in Kirrel1-KO mice after DDC diet? This would be nice to show in addition to Ctgf to confirm that Kirrel1 deficiency might alter proliferation through a Hippo/YAP-independent mechanism
Are there any changes in Kirrel1-KO liver regeneration aside from DR (e.g. after partial hepatectomy or after CCl4)?

Reviewer #3:

Remarks to the Author:

This manuscript describes the identification of KIRREL1 as an upstream component of the mammalian Hippo signaling pathway. The authors show that loss of KIRREL1 increases YAP/TAZ activity, whereas over-expression of KIRREL1 decreases YAP/TAZ activity. They provide evidence that KIRREL1, a membrane protein, interacts with SAV1 and promotes localization of SAV1 to membranes at points of cell-cell contact. They report that KIRREL1 is a transcriptional target of YAP/TAZ, in line with many observations of negative feedback regulation in the Hippo pathway. They also provide some analysis of requirements for KIRREL1 in cancer cell lines, and in a mouse liver regeneration model. It's a substantial contribution to the field, and the analysis appears solid. I think it could be published without any need for additional experiments. Nonetheless, I have a couple suggestions for modifications to the text.

1) At one point, the authors write "these findings suggest that KIRREL1 promotes recruitment of SAV1 to cell-cell contact sites to activate the downstream kinase cassette, linking contact inhibition to the activation of the Hippo pathway."

This suggestion relies on a common mis-understanding of contact inhibition. "Contact inhibition" of cell proliferation is misnamed, because (at least in most cell lines), it's a function of cell density, not simply cell contact. Similarly, the "contact inhibition" regulation of Hippo signaling that's been described in the literature is a function of cell density, not cell contact. The authors should withdraw this suggestion, or else perform additional experiments to demonstrate regulation of Hippo signaling that depends upon KIRREL1 and cell contact, rather than on cell density.

Similarly, the authors note that the effects of KIRREL1 knockout are more evident at high cell density than at low cell density, and suggest therefore that " KIRREL1 might serve as a physical sensor for cell-cell contact-mediated activation of the Hippo pathway to control cell proliferation." However, it can be more simply explained by the fact that Hippo signaling is regulated by cell density. Since knockout of KIRREL1 increases YAP activity, its effects will naturally be more evident when cell density is high, and YAP activity is low, than when cell density is low, and YAP activity is already high. The authors should be careful not to over-interpret these experiments.

2) I would like to see the authors devote more of the Discussion to comparisons between KIRREL1 and *Drosophila* echinoid. To my understanding, echinoid appears to be both a structural and functional

homolog of KIRREL1. It's role in Drosophila Hippo signaling is essentially identical to that described here for KIRREL1, and like KIRREL1 it's an Ig-domain containing transmembrane protein that participates in homophilic adhesion. A simple Blast search shows that echinoid and KIRREL1 have significant sequence similarity, even if other Drosophila proteins in the Ig family score higher. Even if the authors think KIRREL1 is not an echinoid homolog, it's worth discussing more clearly what's similar or different about them in terms of their structure as well as their role in the Hippo pathway.

Reviewer #1 (Remarks to the Author):

In this manuscript, Atanu et al identify KIRREL1 as an upstream regulator of Hippo signaling that binds to SAV1. The authors identify the interaction domains in KIRREL1 and SAV1 and show that KIRREL1 expression is induced upon YAP activation and acts as negative feedback for Hippo signaling. The manuscript is clearly written, and the data presented underscores most of the claims and provides new insight for the YAP research community. The in vivo data presented that underlines the relevance of their findings is not so convincing. The cancer cell lines show indeed a strong correlation of KIRREL1 with known Hippo signaling. However, authors could provide evidence on human liver biopsies that shows that there is a correlation between KIRREL1 and YAP downstream targets. In livers of alcoholic hepatitis for instance in which a strong DR takes place, one would expect high KIRREL1 levels in the expanding BECs. Please find below some additional comments that could further improve the manuscript.

Figure 1.

The screen could be explained better in the M&M; “HEK293A GTIIC-GFPPEST Cas9 cells were transduced with gRNA lentiviral library at MOI 0.5. Fourteen days post lentiviral transduction, cells were grown to high cell density, and sorted using BD FACS Aria Cell Sorter”. Were cells passaged in between, were they selected?

We thank the reviewer for the suggestion. We have now provided a detailed description of the CRISPR screen in Methods, including how cells were infected and passaged (line 472-484).

The raw FACS data used to sort the GFP-High and Low population should be shown? Not just a schematic representation. How was the sort done, on FSC or SSC vs GFP? Were there clear populations visible? This diagram does not give us much information..

As per the reviewer's suggestion, we have included the raw FACS data used to sort the GFP-high and the GFP-low population (Fig. S1c). FSC and SSC were used to identify viable single cells. Since gRNAs were cloned into a viral vector expressing RFP-T2A-Puro, RFP positive cells were sorted into GFP-high and GFP-low populations. Note that the majority of cells were RFP positive as cells were selected with Puromycin. As seen in Fig S1c, a clear GFP-high cell population was visible.

c.

Since high and low GFP populations are sorted, the graph in Fig1b should also show the gRNAs overrepresented in the GFP-low populations for instance (or the top 10 could be given in a table in sup). These enhanced gRNAs should be enhancers of Hippo signaling. It would strengthen the paper to show that such a LOF screen could also identify known enhancers of Hippo.

Our CRISPR screen was designed to identify negative regulators of YAP/TAZ by using a YAP/TAZ transcription reporter as read out. To increase the chance of identifying negative regulators of YAP/TAZ signaling, we purposely let the culture to reach the high cell density (as described in the main text and Methods) on the day of cell harvesting. Since YAP/TAZ are mostly inactive at the high cell density, we had minimal chance to identify positive regulator of YAP/TAZ signaling. Nevertheless, as per the reviewer's request, we have provided the list of top 10 genes enriched in the GFP-low population in the table below for review's eyes. Note that PTK2 is a known positive regulator of YAP/TAZ. YAP and TAZ did not score here, which is possibly due to their functional redundancy in HEK293A cells.

Gene	RSA Down
GET4	-7.18066
GANAB	-6.78927
PTK2	-6.71613
MLLT4	-6.22053
STRAP	-5.99479
DPAGT1	-5.98511
METTL12	-5.95766
WRB	-5.86896
CARNS1	-5.85781
ZDHHC5	-5.73403

KIRREL1 gRNA is used in 1C, but the efficiency is not shown, while this is done for the shRNA of KIRREL (in Figure S4).

We thank the reviewers for raising this issue. We tested many commercial KIRREL1 antibodies and failed to identify a KIRREL1 antibody that can detect endogenous KIRREL1 protein. For this reason, we had to rely on KIRREL1 qRT-PCR assay. Although KIRREL1 shRNAs strongly decreased *KIRREL1* mRNA level, KIRREL1 gRNA modestly decreased *KIRREL1* mRNA (now shown as Fig. S1d). We believe decreased *KIRREL1* mRNA expression is due to nonsense-mediated decay, which targets mRNAs harboring premature termination codon for degradation.

d.

“Control and KIRREL1 knockout cells were plated at high or medium density and subjected to western blot analysis.” What does this mean? Are cells plated at the same density and then grown until medium or high density or immediately seeded at high or medium density? Since the low, medium and high density culture is used several times, images could be shown to illustrate the density.

We thank the reviewer for the suggestion. The cells were plated at low (0.25×10^6 cells/well), medium (0.50×10^6 cells/well) or high (1.0×10^6 cells/well) density in 12-well plate. For Western blot analysis of YAP and TAZ expression, we lysed the cells 24 hours post cell plating. For YAP/TAZ target gene expression and GTIIC-luciferase reporter activity, we performed the assays 48 hours post cell plating. As per the reviewer's suggestion, we have shown images of low, medium, and high density of HEK293A reporter cell line 24 hours post cell plating (Fig. S1a). We have also included such details in figure legends and Methods.

a.

In 1e, it would be good to have a low density control blot next to these medium/high density blots and to show YAP as well. Is KIRREL only affecting TAZ in these conditions?

We thank the reviewer for the suggestion. In revised Fig. 1e, we have included the low density control blot next to the high and medium density blots. In addition, we have also blotted for phospho-YAP and YAP. Although total YAP was not affected upon KIRREL1 KO, phospho-YAP was decreased in *KIRREL1* KO cells at both high and medium densities. Note that phospho-YAP and total TAZ levels were not affected by *KIRREL1* KO at low density.

For f and g, it is not clear at what density the cells were taken for FACS or target gene determination, please specify because density does matter for all the experiments.

For all our FACS assay and qRT-PCR assay, cells were plated at medium density (0.50X10⁶ cells/well) in 12-well plate and collected 48 hours post cell seeding. We have updated the figure legends to reflect cell density and time of cell collection.

Figure 2.

Clearly shows that overexpression of KIRREL affects the expression of TAZ, and decreases target expression of YAP/TAZ downstream genes. Authors do show that overexpressed TAZ-V5 is affected. It would be more convincing to show that endogenous YAP and TAZ is affected by overexpressed KIRREL. This would be more in line with the

target gene analysis and the luciferase assays shown in the rest of the figure, in which no TAZ is overexpressed.

Following reviewer's suggestion, we replaced the old Fig. 2a with this updated figure showing the effect of KIRREL1 overexpression on endogenous YAP and TAZ proteins. Consistent with our initial data, phospho-YAP was increased and total TAZ was decreased upon KIRREL1 overexpression.

The quadruplicate experiments for luciferase, are they biological repeats or technical repeats of one transfection experiment? The M&M states "the experiment was performed in quadruplicates" which seems to suggest that the experiment was done only once? In the figures it would be good to mention "GTIIC-luciferase" (or anything else indicating which luciferase it is) above the luciferase graphs.

The same luciferase assay was performed in four biological replicates for at least three times. We have obtained consistent results in our all luciferase assays as seen in Fig. 2e, Fig. S2b, and Fig. 3g. In addition, we have now included Fig. S3b, which is from an independent experiment, to corroborate findings of Fig. 3g. As per the reviewer's suggestion, we have included details of the assay setup in the Methods section. We have added "GTIIC-Luciferase" above the luciferase graphs.

Figure 3

Clear data showing that the c-term region of the ICD of KIRREL is necessary for the binding with the N-term of SAV-1 and that deletion of this ICD domain affects the luciferase assay. Does it also affect the gene expression of YAP/TAZ downstream genes in a similar way? How about stability of the endogenous YAP/TAZ proteins when this ICD3 mutant is overexpressed?

Following reviewer's suggestion, we have performed qRT-PCR assays to detect changes in YAP/TAZ downstream gene expression in cells overexpressing WT, ICD Δ 1, or ICD Δ 3 (Fig. S3c). As shown in Fig. S3c, overexpression of WT KIRREL1 and ICD Δ 1 mutant, but not ICD Δ 3 mutant, decreased expression of *CTGF*, *CYR61*, and *ANKRD1* mRNA.

Consistent with qRT-PCR and GTIIC-Luc reporter assay data, we have also shown that overexpression of WT KIRREL1 and ICD $\Delta 1$ mutant, but not ICD $\Delta 3$ mutant, decreased the expression of endogenous TAZ protein (Fig. S3d). All these results support the importance of the C-terminus of KIRREL1 in regulation of Hippo signaling.

Figure 4,
The FACS plots show that there seems to be more cells with a higher GFP signal, but there is no clear shift of low to high expressing cells. Can the authors explain why this is. Figure 4S shows that the levels of both KIRREL and SAV are very low, still this does not seem to affect all cells as evidenced by the plots? Is this due to density in the dishes? Authors could discuss this minor affect of knocking down KIRREL and SAV on the transgene.

Within the same cell population, the expression of GTIIC-GFP of a given cell is somewhat stochastic, which might be the reason that the curve of GTIIC-GFP can be broad at certain cell densities. The nature of the assay is very much cell density dependent. While we strived to keep the same experimental condition, the system does fluctuate a bit depending on the cell growth condition/cell density at the time of cell harvesting. Please compare the shape of GTIIC-GFP curve in Fig. 1c, Fig. 1f, and Fig. 4a. In all these experiments, *KIRREL1* knockdown or knockout increased the expression of GTIIC-GFP, although the shape of curve and the degree of right shift can be slightly different.

In 4b. authors claim that there is no significant difference between single and double knock-down, but significance is not given. One could indicate NS between the single and double knock-down bars (if this is indeed the case).

Following reviewer’s suggestion, we have done the statistical significance analysis between SAV1 inhibition and SAV1-KIRREL1 dual inhibition and included it in the revised figure. For *CYR61*, there is no significance. For *CTGF*, there is weak significance (p value 0.0429).

b.

In 4e one shows that KIRREL expression is correlated with density. It would be nice to know what density we are talking about (images of the cells in culture) and to see mRNA levels of targets of YAP (ANKRD1, CTGF etc) alongside KIRREL1 expression. Is the expression really correlated or are there waves of KIRREL1 and YAP-targets?

We assume that the reviewer referred to Fig. 4f (instead of Fig. 4e) where we tested *KIRREL1* expression at different cell densities. We used high (1.0×10^6 cells/well), medium (0.50×10^6 cells/well) or low (0.25×10^6 cells/well) density in 12-well plate and analyzed *KIRREL1* expression by qRT-PCR at 48 hours following cell seeding. We have now indicated the cell density in the figure legends. In addition, as per the reviewer’s suggestion, we have included other YAP target genes – *CTGF*, *CYR61*, and *ANKRD1* in our analysis and replaced our previous figure with this updated figure (Fig. 4f).

Figure 5: “For generating KIRREL1 knockout cancer lines, cancer cell lines were

transduced with lentiviral vectors expressing both Cas9 and gRNA from a single vector and selected with Puromycin.” Please indicate the vector used.

We have now included the description of the vector in Methods (line 447-449).

Figure 6.

This part shows that there is a greater DR when Kirrel1 is deleted by the AlbCre mouse. There is more YAP activation as evidenced by Ctgf ISHs and there is an increased number of CK19+ BECS and SOX9 positive periportal hepatocytes. Wouldn't KO of Kirrel in epithelial cells lead to a higher YAP activity during development, from the start of the AlbCre and thus affect liver growth or size? Authors do comment on this by suggesting the existence of redundant mechanisms to promote SAV1 membrane targeting in hepatocytes. But why are these mechanisms not in place in a the DDC injury model?

We have now performed a detailed analysis of Kirrel1:AlbCre mice and control littermates during postnatal development. As detailed in the Figure for the reviewer, we did not observe any differences in cell fate decisions, proliferation of metabolic zonation in these mice. While our whole liver DNA analysis suggests successful deletion of Kirrel1 at postnatal day (P)10, we cannot exclude the possibility that Kirrel1 is important for liver development. It is known that AlbCre-mediated deletion occurs rather late during embryonic liver development and is incomplete at birth (Postic et al Genesis 2000 PMID10686614). Future studies using a different Cre line will be necessary to assess the role of Kirrel1 during liver development. However, our detailed analyses exclude the possibility that developmental effects biased the observations we made during liver regeneration. We believe that our data addresses this valid point raised by the reviewer but prefer to not include the analysis, provided for the reviewer below, in the manuscript. We discuss the limitations of our mouse model for studying the role of Kirrel1 during liver development in the revised discussion section (line 423-427).

Postnatal liver development in Kirrel1:AlbCre mice. (a, b) Immunofluorescence staining for biliary cell markers (SOX9, PanCK), hepatocyte markers (HNF4 α , ALB), zonation (GS) and proliferation (Ki67) markers (a) and their quantification (b) suggests no differences between Kirrel1:AlbCre and control litter mates during early postnatal liver development (P2 and P10, postnatal days 2 and 10). **(c)** Gene expression analysis using whole liver lysates for YAP target genes in suggests no increased YAP signaling in P2 or P10 Kirrel1:AlbCre mice when compared to control littermates. **(d)** No difference in liver to body weight ratios in P2 or P10 Kirrel1:AlbCre mice when compared to control littermates. **(e)** PCR on whole liver DNA indicates deletion of Kirrel1 in P10 Kirrel1:AlbCre mice. Data presents mean \pm SD. Ns, not significant. Scale bars are 50 μ m (a).

Although our data suggests KIRREL1 positively regulates Hippo signaling through promoting membrane targeting of SAV1, we would stress that KIRREL1 is not equivalent to SAV1 as Kirrel1 liver KO does not have the liver overgrowth phenotype associated with Sav1 KO mice. As seen in Fig.6a, Kirrel1 expression is limited to a subset of CK19+ biliary epithelial cells (BECs) and a small fraction of periportal hepatocytes with no expression around the central vein. This expression pattern is consistent with our finding that Kirrel1 is a YAP target gene. Unlike Kirrel1, Sav1 is broadly expressed in the liver. Restricted expression of Kirrel1 can explain the absence of liver overgrowth phenotype.

We hypothesize that other membrane proteins might recruit Sav1 to the plasma membrane in hepatocytes with low YAP signaling. In DDC injury model, YAP activity is markedly increased in BECs and periportal hepatocytes, which leads to further increase of Kirrel1 expression in these cells. This is likely the reason that Kirrel1 plays a dominant role over other mechanisms in DDC model. We have this point more clear in Discussion (line 420-423).

One aspect that is not shown is whether the injury induced by the 16days DDC in the AlbCRE Kirrel mice was different? If the absence of Kirrel in hepatocytes and BECs leads to more hepatocyte cell death due to this diet, than the DR would be greater as well. Authors should show ALT/AST, Billirubin etc. measurements and for instance liver to bodyweight ratios, showing that there is no difference in liver damage.

The reviewer raised a very important point since increased injury and inflammation may have contributed to the increased DR we observed in Kirrel1:AlbCre mice. We have now performed the suggested analyses and in addition assessed the degree of inflammation in Kirrel1:AlbCre mice and control littermates post 16 days DDC injury. Our detailed analyses suggest comparable injury and inflammatory damage response in both groups, excluding that a different degree of injury may have biased our DR assessment. These results are now shown in Fig. S7a-d.

However, authors could provide evidence on human liver biopsies that shows that there is a correlation between KIRREL1 and YAP downstream targets in livers of alcoholic hepatitis for instance in which a strong DR takes place, one would expect high KIRREL1 levels in the expanding BECs.

We thank the reviewer for suggesting this elegant experiment. We have obtained human liver biopsies from patients with an injury-induced DR and established ISH stainings for *KIRREL1* and the YAP target gene *KLF6*. Within each patient, there was a clear difference in *KIRREL1* and *KLF6* expression in areas of a DR (high expression in expanding BECs) compared to the liver parenchyma (low expression in hepatocytes distant to the portal vein). Importantly, we found a very high correlation between *KIRREL1* and *KLF6* ISH signals, suggesting that *KIRREL1* expression correlated with YAP signaling activity also during a DR in patients. These results are now shown in Fig. S7e-f.

Minor comments:

Line 40: Zhao 2007 is used instead of a number

Line 78: CRISPR screens or CRISPR screening iso screen

Line 252: (Fig. 5c) should be Fig. 5d.

Line:256 Fig. 5d should be Fig. 5e

Line:259: Fig. 5e should be Fig. 5f

These mistakes have been corrected in the revised manuscript. We thank the reviewer for pointing them out.

Reviewer #2 (Remarks to the Author):

In an elegant approach combining genome-wide CRISPR screen with a reporter of YAP/TEAD activity, Paul et al. identified KIRREL1 as a novel upstream regulator of the Hippo signaling cascade. Using loss-of function and overexpression experiments, the authors convincingly show that KIRREL1 negatively regulates YAP/TAZ activity through binding to the adaptor protein SAV1 – an interaction that seems to be specifically relevant at cell-cell contact and might even influence YAP/TEAD activity in neighboring cells. Supporting evidence for the functional relevance of KIRREL1 in the control of Hippo/YAP signaling is provided by analysis publicly available screening data from different cancer cell lines and more importantly, from in vivo data in mice with liver-specific Kirrel1 deletion.

Overall, the manuscript is well written and the data presented support the authors' conclusions. With identification of a completely novel regulator of Hippo/YAP signaling,

the results are of high relevance in the field However, some parts of the manuscript outlined below should be addressed:

1) Figure 1e: only TAZ levels are shown. What happens to YAP expression and YAP/TAZ phosphorylation? If changes in YAP levels cannot be observed in HEK293 cells this should be addressed.

Reviewer #1 also raised the same point. In the updated Fig. 1e, we have shown that *KIRREL1* KO decreased phosphorylation of YAP, but not the total level of YAP, at the medium and high cell densities. *KIRREL1* KO also increased TAZ level at the medium and high cell densities. These findings are consistent with the notion that Hippo signaling mainly regulates YAP at the nuclear translocation level and regulates TAZ at the protein degradation level.

2) Figure 4c: SAV1 and KIRREL also seem to co-localize the nuclear membrane. Is this observed on a regular basis? What could be the function at this nuclear interaction as it should not be relevant for sensing of cell-cell contacts?

The reviewer is correct; we do see SAV1 and KIRREL1 co-localization at the nuclear membrane. However, we believe this is an artifact associated with KIRREL1 overexpression. Endoplasmic Reticulum (ER)-Golgi-mediated vesicular transport ensures proper membrane targeting of the plasma membrane proteins. KIRREL1 is a single-pass cell surface protein with a signal peptide at its N-terminus. A small portion of overexpressed KIRREL1 is localized on the membrane of ER with the intracellular domain (ICD) exposed to the cytosol. It is likely that SAV1 binds to the ICD of overexpressed KIRREL1 during the vesicular transport of KIRREL1 through the ER. Because the outer nuclear membrane is continuous with the membrane of the ER, it may seem that SAV1 and KIRREL1 co-localize on the nuclear membrane.

3) Figure 4d: how is the difference in reporter GFP expression in RFP- vs RFP+ cells? This would help to show how important the influence of KIRREL1 is on neighboring cells in comparison the cell autonomous effects. Immunofluorescence staining for YAP would help to strengthen the results of this otherwise very elegant experiment.

The reviewer suggested us to compare GTIIC-GFP in parental RFP- cells and *KIRREL1* KO RFP+ cells to see the cell autonomous effect of *KIRREL1* KO. Although this is a good idea, there is a caveat for this comparison. As seen in figure below, we have plotted GTIIC-GFP signal in parental HEK293 GTIIC-GFP cells (RFP-) cells and HEK293 GTIIC-GFP cells infected lentivirus expressing control gRNA (RFP+) cells from the same experiment shown in Fig. 4d. In principle, GTIIC-GFP signal should be the same in these two populations as control gRNA is not supposed to affect YAP activity. However, we found increased GTIIC-GFP signal in RFP+ cells.

We believe that viral infection affects cell growth/behavior, which indirectly influences GTIIC-GFP signal. For this reason, the comparison suggested by the reviewer would not be meaningful. Cell autonomous effect of KIRREL1 KO can be seen in Fig 1c, where HEK293A GTIIC-GFP Cas9 cells transduced with lentivirus expressing RFP and control or KIRREL1 gRNAs were mixed with parental cells at 1:3 ratio, co-cultured for 2 days, and subjected to FACS analysis. Experiments similar to Fig. 1c have been performed multiple times in our hands. Based on our experience, the degree right shift of GTIIC-GFP seen in Fig. 4d (non-cell autonomous effect) is comparable to what we see in experiments similar to Fig. 1c (cell autonomous effect).

We are glad that the reviewer appreciate the significance of our co-culture data. While we tried immunostaining for YAP, we encountered significant technical difficulties. Since the red (555 nm) and green (488 nm) channels were already used for RFP and GFP, respectively, we had to use far-red channel (647 nm) to stain YAP in these cells. However, the signal and the resolution of YAP staining in the far-red channel were very poor, even after several optimization experiments. Therefore, we could not detect clear YAP signal and quantify the nuclear/cytoplasmic YAP staining as suggested by the reviewer. Nevertheless, we feel that the conclusion of our Fig. 4d is solid. We think GTIIC-GFP is a reliable readout of YAP activity; through our study, we have seen a nice correlation between GTIIC reporter and the expression of endogenous YAP target genes. In addition, one can measure GTIIC-GFP in many cells and obtain unbiased data.

4) Figure 5e: if possible, the authors should show YAP/TAZ protein levels or immunostaining for YAP in at least some of these cell lines

As per the reviewer's suggestion, we have included Western blot analysis of YAP, TAZ, and phospho-YAP in the IGR-39 cell line (Fig. S5f). This cell line showed significant increase of cell proliferation upon *KIRREL1* KO. As seen in Fig. S5f, *KIRREL1* KO decreased YAP phosphorylation and increased TAZ protein expression, suggesting that *KIRREL1* KO activates YAP/TAZ signaling in this cell line.

f.

Quantifying YAP nuclear staining can be somewhat subjective. Instead of performing YAP immunostaining, we performed YAP nuclear fractionation in HEK293A cells to corroborate our findings. As shown in Fig. S1e, *KIRREL1* KO increased nuclear YAP in HEK293A cells.

e.

5) Figure 6: are there any changes in hepatic YAP localization by IHC in *Kirrel1*-KO mice after DDC diet? This would be nice to show in addition to *Ctgf* to confirm that *Kirrel1* deficiency might alter proliferation through a Hippo/YAP-independent mechanism Are there any changes in *Kirrel1*-KO liver regeneration aside from DR (e.g. after partial hepatectomy or after CCl4)?

We have tested several YAP antibodies and received protocols for a commercial YAP antibody from Michael Dill, a former postdoc in Fernando Camargo's lab. Unfortunately, the batch we ordered did neither yield reliable staining (we tested it on YAP-KO mice) nor did we see correct subcellular localization (mouse and human livers with activated YAP signaling). We had similar problems with other commercial antibodies and no access to lab-made ones from other YAP labs. For the new human data generated in response to reviewer 1, we therefore used another YAP target gene (*KLF6*) to correlate *KIRREL1* expression with YAP signaling activity. Regarding the point the reviewer made about additional regeneration models, we have selected the DDC model since it induces a strong DR, the major regenerative response in which YAP plays a role (but WNT signaling does not; see Planas-Paz and Sun et al., Cell Stem Cell 2019). Following partial hepatectomy and CCL4 damage of zone3 hepatocytes, it is mainly other

pathways (like e.g, WNT signaling) controlling the regenerative response. In fact, the Halder lab has recently shown that YAP signaling is mostly dispensable for liver regeneration in partial hepatectomy and CCL4 models (Verboven et al, Gastroenterology 2021, PMID: 33127392). In line with their findings, we also did not find major defects in hepatocyte-mediated liver regeneration in YAP1:AlbCre mice (data not published) and hence did not perform the additional regeneration experiments. We rather believe that YAP and Kirrel1 are critical for regeneration during a DR and therefore kept the focus of our investigation on this YAP-driven regenerative process, that we expanded with new human data (see response to reviewer 1). To reflect this, we have changed the title of the result section from "KIRREL1 serves as a feedback regulator restricting YAP/TAZ activity during liver regeneration" to "KIRREL1 serves as a feedback regulator restricting YAP/TAZ activity during ductular reaction" (line 298).

Reviewer #3 (Remarks to the Author):

This manuscript describes the identification of KIRREL1 as an upstream component of the mammalian Hippo signaling pathway. The authors show that loss of KIRREL1 increases YAP/TAZ activity, whereas over-expression of KIRREL1 decreases YAP/TAZ activity. They provide evidence that KIRREL1, a membrane protein, interacts with SAV1 and promotes localization of SAV1 to membranes at points of cell-cell contact. They report that KIRREL1 is a transcriptional target of YAP/TAZ, in line with many observations of negative feedback regulation in the Hippo pathway. They also provide some analysis of requirements for KIRREL1 in cancer cell lines, and in a mouse liver regeneration model.

It's a substantial contribution to the field, and the analysis appears solid. I think it could be published without any need for additional experiments. Nonetheless, I have a couple suggestions for modifications to the text.

1) At one point, the authors write "these findings suggest that KIRREL1 promotes recruitment of SAV1 to cell-cell contact sites to activate the downstream kinase cassette, linking contact inhibition to the activation of the Hippo pathway."

This suggestion relies on a common mis-understanding of contact inhibition. "Contact inhibition" of cell proliferation is misnamed, because (at least in most cell lines), it's a function of cell density, not simply cell contact. Similarly, the "contact inhibition" regulation of Hippo signaling that's been described in the literature is a function of cell density, not cell contact. The authors should withdraw this suggestion, or else perform additional experiments to demonstrate regulation of Hippo signaling that depends upon KIRREL1 and cell contact, rather than on cell density.

Similarly, the authors note that the effects of KIRREL1 knockout are more evident at high cell density than at low cell density, and suggest therefore that " KIRREL1 might serve as a physical sensor for cell-cell contact-mediated activation of the Hippo pathway to control cell proliferation." However, it can be more simply explained by the fact that Hippo signaling is regulated by cell density. Since knockout of KIRREL1 increases YAP activity, its effects will naturally be more evident when cell density is high, and YAP activity is low, than when cell density is low, and YAP activity is already high. The authors should be careful not to over-interpret these experiments.

We thank the reviewer for these nice comments/suggestions. We agree with the reviewer that Hippo signaling is regulated by cell density, and not just cell contact. Cell contact is likely one of several factors involved in density-regulated Hippo signaling. On the other hand, we do think KIRREL1 is likely involved in cell-cell interaction-mediated YAP regulation. As shown in Fig. 4d, KIRREL1 KO increases YAP activity in neighboring cells harboring wild-type KIRREL. In our mind, this finding cannot be easily explained by simple cell crowding. However, firmly demonstrating that the effect of KIRREL1 is mediated by cell contact and not by cell density is not possible as these two processes are closely intertwined. In addition, the strength of KIRREL1-KIRREL1 trans-interaction could be different at different cell densities. Therefore, we have removed the term “contact inhibition” from the manuscript. We have weakened our language to “these findings suggest that KIRREL1 promotes recruitment of SAV1 to cell-cell contact sites to activate the downstream kinase cassette, linking high cell density to the activation of the Hippo pathway”. Similarly, we have weakened the original statement “KIRREL1 might serve as a physical sensor for cell-cell contact-mediated activation of the Hippo pathway to control cell proliferation” to “KIRREL1 might serve as a sensor for cell-cell contact in high cell density-mediated activation of the Hippo pathway and inhibition of cell proliferation”. We hope these modifications will satisfy the reviewer.

2) I would like to see the authors devote more of the Discussion to comparisons between KIRREL1 and Drosophila echinoid. To my understanding, echinoid appears to be both a structural and functional homolog of KIRREL1. It's role in Drosophila Hippo signaling is essentially identical to that described here for KIRREL1, and like KIRREL1 it's an Ig-domain containing transmembrane protein that participates in homophilic adhesion. A simple Blast search shows that echinoid and KIRREL1 have significant sequence similarity, even if other Drosophila proteins in the Ig family score higher. Even if the authors think KIRREL1 is not an echinoid homolog, it's worth discussing more clearly what's similar or different about them in terms of their structure as well as their role in the Hippo pathway.

We thank the reviewer for the suggestion and we have now included one paragraph around the similarity between Echinoid and KIRREL1 in the Discussion section (line 386-393). Indeed, the structure and the function of KIRREL1 and Echinoid in Hippo signaling are similar. Although other Drosophila Ig family proteins might have higher KIRREL1 homology as compared with Echinoid, KIRREL1 appears to be a functional mammalian homolog of Echinoid.

Reviewers' Comments:

Reviewer #1:

Remarks to the Author:

I very much appreciate the detailed answers to the concerns and questions raised. I value the changes and believe that this is an important study that gives us some more insight into the interesting world of YAP.

Reviewer #2:

Remarks to the Author:

The authors provided experiments to address the concerns raised by the reviewers. If this was not possible, a detailed clarification of the underlying reasons why that did not perform a requested experiment was provided in the rebuttal. Except for one minor concern regarding new Figure S7e, the paper can be recommended for publication.

New Fig S7e: the data presented are very nice, but from IF for CK19 it is hard to get a good overview of the ductular reaction presented in this slide (DR in human livers are often less clear than in murine models). If available, it would be nice to have H&E staining and/or IHC for CK19 of the same area.

Reviewer #3:

Remarks to the Author:

I am fully satisfied by the authors revisions. As noted before, I think it's a substantial contribution.

Reviewer #2

New Fig S7e: the data presented are very nice, but from IF for CK19 it is hard to get a good overview of the ductular reaction presented in this slide (DR in human livers are often less clear than in murine models). If available, it would be nice to have H&E staining and/or IHC for CK19 of the same area.

Providing a CK19 IHC staining for the exact same region would be limited by the possibility to find the exact same region again. It would further require fresh patient samples from our collaborator and repetition of the ISH staining, as the signal on the stained slides decayed by now and does not offer nice representative images anymore, correlating with our quantification. Instead, we now extracted the CK19 channel from the original image and show it separately in Figure S7e, to guide the reader on where the ductular reaction is. We believe that the amended Figure is now clearly showing the ductular reaction and parenchymal regions with KIRREL/KLF6 signals in the human liver.